# Predicting vaccine effectiveness against severe COVID-19 over time and against variants: a meta-analysis

Deborah Cromer [1] ✉, Megan Steain[2,3], Arnold Reynaldi [1], Timothy E. Schlub[1,4], Shanchita R. Khan [1], Sarah C. Sasson[1], Stephen J. Kent [5,6], David S. Khoury [1] & Miles P. Davenport [1]

Vaccine protection from symptomatic SARS-CoV-2 infection has been shown to be strongly correlated with neutralising antibody titres; however, this has not yet been demonstrated for severe COVID-19. To explore whether this relationship also holds for severe COVID-19, we performed a systematic search for studies reporting on protection against different SARS-CoV-2 clinical endpoints and extracted data from 15 studies. Since matched neutralising antibody titres were not available, we used the vaccine regimen, time since vaccination and variant of concern to predict corresponding neutralising antibody titres. We then compared the observed vaccine effectiveness reported in these studies to the protection predicted by a previously published model of the relationship between neutralising antibody titre and vaccine effectiveness against severe COVID-19. We find that predicted neutralising antibody titres are strongly correlated with observed vaccine effectiveness against symptomatic (Spearman $\rho$ = 0.95, $p$ < 0.001) and severe (Spearman $\rho$ = 0.72, $p$ < 0.001 for both) COVID-19 and that the loss of neutralising antibodies over time and to new variants are strongly predictive of observed vaccine protection against severe COVID-19.

Immunisation against SARS-CoV-2 has been shown to be highly effective in preventing both mild and severe COVID-19[1–3]. Previous work has demonstrated that neutralising antibody responses are highly predictive of vaccine efficacy against symptomatic SARS-CoV-2 infection[4–7]. However, the waning of antibody titres and the emergence of SARS-CoV-2 variants with an immune escape from vaccine-induced neutralising antibodies has contributed to declining vaccine efficacy against symptomatic infection[8–12]. Despite the loss of vaccine efficacy against symptomatic infection, protection against hospitalisation and death has remained high, leading some to speculate on an apparent 'decoupling' of the mechanisms of protection against mild and severe COVID-19[13].

Studies of the relationship between neutralising antibodies and protection from symptomatic SARS-CoV-2 infection have shown that neutralising antibody titres are highly predictive of vaccine efficacy[4–7,14,15] and that a titre of ~20% of the early convalescent level is associated with 50% protection from symptomatic infection[4]. This relationship predicted that a decline in neutralising antibody titres, either as a result of waning immunity or changes in SARS-CoV-2 viral variants, will lead to reduced vaccine protection against COVID-19[4].

[1]Kirby Institute, University of New South Wales, Sydney, Australia. [2]Sydney Institute of Infectious Diseases and Charles Perkins Centre, Faculty of Medicine and Health, The University of Sydney, Sydney, Australia. [3]School of Medical Sciences, Faculty of Medicine and Health, The University of Sydney, Sydney, Australia. [4]Sydney School of Public Health, Faculty of Medicine and Health, University of Sydney, Sydney, Australia. [5]Department of Microbiology and Immunology, University of Melbourne at the Peter Doherty Institute for Infection and Immunity, Melbourne, Australia. [6]Melbourne Sexual Health Centre and Department of Infectious Diseases, Alfred Hospital and Central Clinical School, Monash University, Melbourne, Australia. ✉e-mail: d.cromer@unsw.edu.au

Analysis of protection against SARS-CoV-2 variants suggests that neutralising antibody levels remain predictive of protection against the Alpha, Beta, Delta and Omicron variants[16,17]. It was also demonstrated that efficacy against severe outcomes was associated with a lower neutralising antibody titre, and it was estimated that the 50% protective titre against severe COVID-19 is approximately 6.5-fold lower than against symptomatic infection[4]. However, this finding relied on data from five phase 3 studies with a combined total of only 60 severe cases. Thus, it was not possible to directly demonstrate a correlation between neutralisation titres and protection from severe outcomes (Fig. 1A)[4]. Importantly, this lower level of neutralising antibodies associated with protection from severe COVID-19 (compared to mild disease) directly predicted that protection against severe infection will persist for longer as antibody levels wane and will be better maintained against new variants[4] (Fig. 1A). However, the relationship between neutralising antibody titres and protection from severe COVID-19 has not been definitively tested, largely because of the small number of severe cases observed in the phase 3 vaccine trials, and thus the difficulty in directly relating neutralising antibodies with protection from severe COVID-19. Since a direct analysis of whether neutralising antibodies are correlated with severe outcomes has not been possible to date, we address this question with a novel approach. We aggregated data from multiple observational studies which report on vaccine efficacy and effectiveness over time and against different SARS-CoV-2 variants. We then used previously published estimates of the decay of neutralising antibodies and the loss of neutralisation against each variant to predict neutralising antibodies for each vaccine, variant and time combination. Using a previously published relationship between neutralising antibodies and protection, we test whether knowing the vaccine regimen, time since vaccination, and SARS-CoV-2 variant, allows us to predict vaccine effectiveness against severe COVID-19. This will provide evidence as to whether changes in neutralising antibodies over time and against different variants are predictive of changes in vaccine effectiveness against severe outcomes.

## Results

### Analysis of vaccine effectiveness in epidemiological studies

To understand vaccine effectiveness against severe SARS-CoV-2 infection, we searched the literature for studies that reported on primary course vaccine effectiveness against symptomatic and severe COVID-19, where results were reported by vaccine, circulating variant(s) and time since vaccination. We identified and extracted data from 15[8–11,18–28] studies that reported vaccine efficacy or effectiveness in this way. These were comprised of two randomised controlled trials, seven test-negative case-control studies (TNCC), and six cohort studies (see supplementary materials and Table 1). These included studies of BNT162b2 (eight studies), mRNA-1273 (six studies), ChAdOx1 nCoV-19

(five studies) and any mRNA vaccine (that aggregated BNT162b2 and mRNA-1273 vaccines; five studies). These studies reported protection against pre-Delta (predominantly ancestral (Wuhan-like) and Alpha, seven studies), Delta (12 studies) and Omicron (four studies) variants. The studies reported vaccine protection against symptomatic infection (nine studies) and severe COVID-19 (10 studies). Several studies reported on more than one variant or vaccine. A summary of the vaccines and variants used in the studies analysed is shown in Fig. 1B–D and given in Table 1.

Figure 2 presents the aggregated data on vaccine effectiveness in preventing severe COVID-19 for different vaccines, variants and time since vaccination. These are key parameters of interest, as they have each been shown to independently influence neutralising antibody levels[4,16,29], and hence may each have independent impacts on protection. In a first analysis of the aggregated data, we used a linear mixed effects model to investigate the impact of vaccine type, variant and time since vaccination on vaccine effectiveness against severe COVID-19 (Eq. 1, Fig. 2 and Supplementary Table S1) while accounting for the potential random variation induced by using data from different studies. This showed that the reported effectiveness against severe COVID-19 did indeed vary by vaccine, variant and over time. For example, vaccination with mRNA-1273 showed higher effectiveness than vaccination with ChAdOx1 nCoV-19 (6.2% higher, 95% CI 3.8–8.6). Similarly, effectiveness against severe COVID-19 was lower against Omicron than against either Delta or the pre-Delta variants (31.4% lower than against the pre-Delta variants, 95% CI 27–35.8). We also found that effectiveness declined over time since vaccination, with a decrease in effectiveness of 1.7% (95% CI 1.4–2.1%) per month. All these findings are qualitatively in line with what would be expected as a result of previously reported changes in neutralising antibody levels for the different vaccines[4], variants[16] and time since vaccination[4,30]. We therefore next consider whether these shifts in vaccine effectiveness against severe outcomes over time for different vaccines and variants are correlated with neutralising antibody titres.

### Correlation between neutralising antibody titre and vaccine effectiveness against severe COVID-19

To understand the relationship between in vitro neutralisation titre and protection, we aggregated data on neutralisation titre and effectiveness across all the studies. The epidemiological studies of vaccine effectiveness did not include contemporaneous measurements of neutralising antibody titres against the different variants. Thus, we asked whether information on vaccine regimen, time since vaccination and circulating variant could be used to predict neutralising antibody titres and if this proxy neutralisation titre was associated with vaccine effectiveness for each reported real-world effectiveness value in the meta-analysis based on the vaccine, variant and time since vaccination.

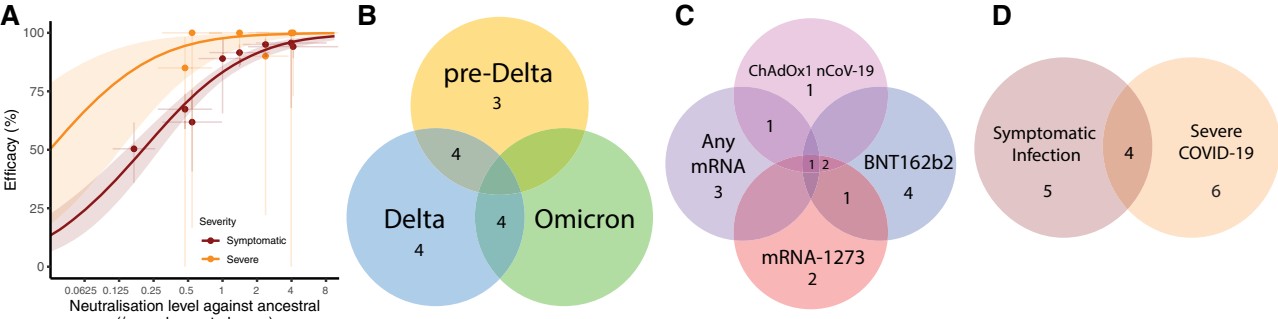

**Fig. 1 | Summary of previously available data linking neutralising antibodies and vaccine effectiveness and data contributing to this analysis. A** Previously reported relationship between neutralising antibody titre and vaccine efficacy in the prevention of symptomatic (dark red) and severe (orange) COVID-19 (reproduced from Khoury et al.[4]). Solid lines indicate best-fit model and shaded areas indicate 95% confidence intervals. Neutralisation titre and efficacy data used to parameterise the model are indicated as dots (95% CI indicated as whiskers). **B–D** Summary of the clinical studies used in this analysis (Table 1).

**Table 1 | Data sources for efficacy data**

| Study | Vaccine | Timeframe postvaccination | Measure/s of effectiveness | Age groups included | Variants analysed | Study type | Country | Data derived from |
|---|---|---|---|---|---|---|---|---|
| Goldberg et al.[25] | BNT162b2[a] | 1.5–6.5 months | 1) PCR positive 2) Severe disease[b] | 16–39 40–59 ≥60 | Delta | Retrospective cohort study | Israel | Table S7 |
| Chemaitelly et al.[10] | BNT162b2 | 0.5 to ≥7 months | 1) PCR positive 2) Hospitalisation (severe disease critical disease and fatal) | any | Alpha, Beta[c], Delta | Test-negative case control | Qatar | Table 2 |
| Keehner et al.[27] | BNT162b2 or mRNA-1273 | 1–7 months | PCR positive and >1 symptom | ≥18 | Delta[d] | Retrospective cohort study | USA | Calculated based on attack rates for July given in text |
| Andrews et al.[9] | BNT162b2 ChAdOx1 nCoV-19 mRNA-1273 | 0.5 to ≥9 months | 1) PCR-confirmed symptomatic disease 2) Hospitalisation 3) Death | ≥16 ≥65 40–64 16–39 | Alpha Delta | Test-negative case control | England | Table 1 Table 2 Table S10 Table S11 Table S12 |
| El Sahly et al.[24] | mRNA-1273 | 0.5 to ≥8 months | 1) Prevention of illness 2) Prevention of severe disease | ≥18 to <65 ≥65 | Ancestral | Randomised controlled trial | USA | Supplementary Table S30 |
| Thomas et al.[20] | BNT162b2 | 0.25 to ≥4 months | 1) Laboratory-confirmed disease (≥1 symptom) | ≥12 | pre-Delta | Randomised controlled trial | USA Argentina Brazil South Africa Germany Turkey | Fig. 2 |
| Rosenberg et al.[26] | BNT162b2 mRNA-1273 | 0–8 months | 1) Laboratory-confirmed disease 2) Hospitalisation | 18–49 50– 5 | pre-Delta Delta | Prospective cohort study | USA | Tables 2 and 3 |
| Andrews et al.[8,47] | BNT162b2 ChAdOx1 nCoV-19 mRNA-1273 | 0.5 to ~25 months | 1) PCR-confirmed symptomatic disease 2) Hospitalisation | ≥18 | Delta Omicron | Test-negative case control | England | Table 3 |
| Ferdinands et al.[22] | mRNA vaccines | 0.5–25 months | 1) PCR-confirmed hospital presentation 2) PCR-confirmed hospital admission | ≥18 | Delta Omicron | Test-negative case control | USA | Table 2 |
| Poukka et al.[28] | mRNA vaccines ChAdOx1 nCoV-19 | 0–8 months | 1) Laboratory-confirmed infection 2) Hospitalisation | 16–69 | pre-Delta Delta | Retrospective cohort study | Finland | Supplementary Tables 2 and 3 |
| Tseng et al.[11] | mRNA-1273 | 0.5–25 months | 1) Infection 2) Hospitalisation | ≥18 | Delta Omicron | Test-negative case control | USA | Table 2 |
| Skowronski et al.[21] | BNT162b2ChAdOx1 nCoV-19 mRNA-1273 | 0.5–11.5 months | 1) PCR-positive infection 2) Hospitalisation | ≥18 | Delta | Test-negative case control | Canada | Supplementary Tables 13 and 14 |
| Thompson et al.[23] | mRNA vaccines | 0.5–25 months | Hospitalisation | ≥18 | Delta Omicron | Test-negative case control | USA | Table 2 |
| Bianchi et al.[18] | BNT162b2 | 0.5–5 months | 1) PCR-confirmed infection 2) Symptomatic disease | 21–70 | pre-Delta | Observational cohort study | Italy | Table 3 |
| Katikireddi et al.[19] | ChAdOx1 nCoV-19 | 0–5 months | 1) Confirmed symptomatic infection 2) Hospital admissions or death | ≥18 18–64 65–79 ≥80[f] | Delta | Retrospective cohort study | Scotland Brazil[e] | Tables 2, 3, Table S19 |

[a]Fully vaccinated = 7 days + post second dose.
[b]Severe disease efficacy included for ages >40 years.
[c]Effectiveness data against Beta infections were not included in this analysis.
[d]Infections were >95% Delta in the timeframe analysed.
[e]Only effectiveness data from Scotland was used, as this was the only cohort that used an unvaccinated reference cohort.
[f]Effectiveness data from a cohort comprised exclusively of individuals over 80 years of age was not included in our analysis.

That is, the expected geometric mean neutralisation titre corresponding to each vaccine effectiveness estimate can be predicted based on: (1) the geometric mean neutralisation titre (GMT) previously estimated for each vaccine[4,16,31], (2) the rate of waning of neutralising antibody levels[4,30], and (3) the drop in neutralisation titre to variants (detailed methods and parameters are given in the "Methods", supplementary materials, Supplementary Tables S3 and S4[4] and Eq. 1 and Eq. S4).

For example, Chemaitelly et al.[10] measured vaccine protection against pre-Delta and Delta variants after vaccination with the BNT162b2 vaccine. This included follow-up for more than 25 weeks after vaccination, with vaccine effectiveness reported for the periods 0–4, 5–9, 10–14, 15–20, 21–25 and ≥25 weeks. The peak GMT titre for BNT162b2 vaccinees against the ancestral variant is estimated to be 2.4-fold (95% CI = 1.5–3.8)[4] of the convalescent GMT. Neutralisation titres against the Delta variant are estimated to be 3.9-fold lower (95% CI = 3.1–4.9)[16] than against the ancestral variant, and neutralising antibody titres have been estimated to wane with a half-life of 108 days (95% CI = 82–159)[4]. Therefore, after accounting for the initial neutralisation titre fold-drop against the variant, decay of antibodies, and the length of the original clinical trials (see "Supplementary Methods"), the GMT of BNT162b2 vaccinees against the Delta variant over the reported periods is estimated to be 0.55 (95% CI 0.33–0.91), 0.46 (95% CI 0.27–0.76) and 0.38 (95% CI 0.22–0.64)-fold of the convalescent GMT against the ancestral virus for 5–9, 10–14, and 15–20 weeks postvaccination, respectively (confidence intervals were obtained by bootstrapping, see supplement). The corresponding real-world vaccine effectiveness estimates of vaccine efficacy against severe COVID-19 caused by the Delta variant in these periods are 100% (95% CI 74.3–100), 81.6% (95% CI 0–99.6) and 100% (95% CI 0–100), respectively (confidence limits as reported in the clinical studies).

The estimates of neutralising antibody titre and data on vaccine effectiveness from the 15 studies included in the meta-analysis were then aggregated and compared. We observe an excellent correlation between the log-10 of the predicted neutralisation titre (x-axis) and reported vaccine effectiveness obtained from our meta-analysis (Spearman's correlation $\rho$=0.95 and 0.72 for symptomatic and severe efficacy respectively, $p < 0.001$ for both, Fig. 3A, B). We explored whether this association might be driven by group differences between study types, vaccines or variants (Supplementary Fig. 1). However, the strong correlation between predicted neutralisation titre and protection from severe SARS-CoV-2 remained across these different subgroupings. We note that this observed correlation between estimates of neutralising antibody titres and effectiveness is independent of the model developed by Khoury et al.[4]. Rather, once published neutralising antibody titres are adjusted to account for (1) immune waning and (2) a drop in recognition of the circulating variant, the reported vaccine effectiveness is remarkably well correlated with these neutralising antibody titres.

### Neutralising antibody titres predict protection from symptomatic and severe COVID-19

The analysis above shows a strong correlation between predicted antibody titres and observed protection against the acquisition of symptomatic and severe COVID-19. However, this does not provide a direct means of predicting vaccine effectiveness based on neutralisation titres. We have previously published a mathematical model relating neutralising antibody titre to vaccine protection (based on the results of the phase 3 trials for seven vaccines and protection seen in convalescent subjects against infection with the ancestral SARS-CoV-2). We therefore next plotted the reported vaccine effectiveness from our meta-analysis against the estimated neutralising antibody titres and compared this to our previously reported relationship between antibody levels and protection (Fig. 3A, B).

Figure 3A shows the previously reported relationship between neutralising antibody titre and protection from symptomatic SARS-CoV-2 infection (red line and shaded 95% confidence intervals, reproduced from ref. [4]). This line is overlaid with the 157 values of estimated neutralisation titre/reported effectiveness combinations we were able to obtain for protection against symptomatic infection from reported studies. Similarly, the 206 reported values for protection from severe COVID-19 are also plotted on top of our previous estimate of the relationship between neutralising antibodies and protection from severe disease (Fig. 3B). We observe that 165/206 (80%) of real-world estimates for vaccine efficacy against severe COVID-19 lie within the confidence intervals of the prediction. Of the estimates that lay outside the confidence intervals, 29/41 (71%) had both reported and predicted efficacies of above 90% (i.e., they were at the very right-hand side of Fig. 3B and Supplementary Fig. S2B where the confidence intervals are very narrow).

Predicted vaccine efficacy and reported vaccine effectiveness are also highly correlated (Pearson $p$-value < 0.001 for both symptomatic and severe infection and $R = 0.93$ and 0.79, respectively. See Supplementary Fig. S2), consistent with the previously reported relationship between neutralisation titre and protection[4]. Similarly, we also generated the mean estimate and 95% confidence intervals for predicted vaccine effectiveness over time across different vaccines, variants and over time for both symptomatic and severe COVID-19 (Supplementary Figs. S3 and S4). Despite the heterogeneity in the epidemiological setting and trial design of the clinical studies, the previously reported model[4] predicting vaccine efficacy based on neutralising antibody titres was in very good agreement with vaccine effectiveness against both symptomatic and severe COVID-19.

### Effectiveness against symptomatic infection predicts effectiveness against severe COVID-19

The analysis above shows that the previously derived relationship between neutralising antibody titres and protection from severe SARS-CoV-2 infection is predictive of the observed protection in observational studies and RCTs (Fig. 3A, B). A potential limitation in this analysis is that neutralisation titres were not directly measured in the epidemiological studies, and thus predictions relied on an estimation of neutralisation titres based on the vaccine type, variant and time since vaccination. Therefore, we next sought to assess the utility of the published model of correlates of protection from severe COVID-19 using an approach that did not rely on estimating neutralising antibody titre. The published correlates model[4] explicitly predicts a (nonlinear) relationship between protection from symptomatic disease and protection from severe disease (red line in Fig. 3C). That is, for any observed level of protection from symptomatic infection, the published model implicitly predicts a corresponding level of protection from severe COVID-19. This has the major advantage of being independent of any assumptions of the underlying neutralising antibody titres. Figure 3C shows the relationship between symptomatic and severe protection predicted from the correlates model (solid line and shaded 95% confidence intervals) and the data from a subset of four studies that reported protection against symptomatic and severe COVID-19 for comparable groups of subjects (Fig. 1D). This subset of data included 198 observational measurements corresponding to 99 matched symptomatic and severe efficacy measurements. The data are shown in Fig. 3C as points and associated 95% confidence intervals (whiskers). We observe that the real-world data points maintain the predicted relationship between symptomatic and severe protection (Spearman's $\rho = 0.7$, $p < 0.001$) and note that 78/99 real-world data points (79%) have a severe efficacy that lies within the 95% confidence intervals based on the corresponding symptomatic efficacy. We note that of the 21 reported points that lie outside of the model confidence intervals, 17 (81%) had predicted efficacies against severe disease above 97%. This is a range where it can be difficult to accurately

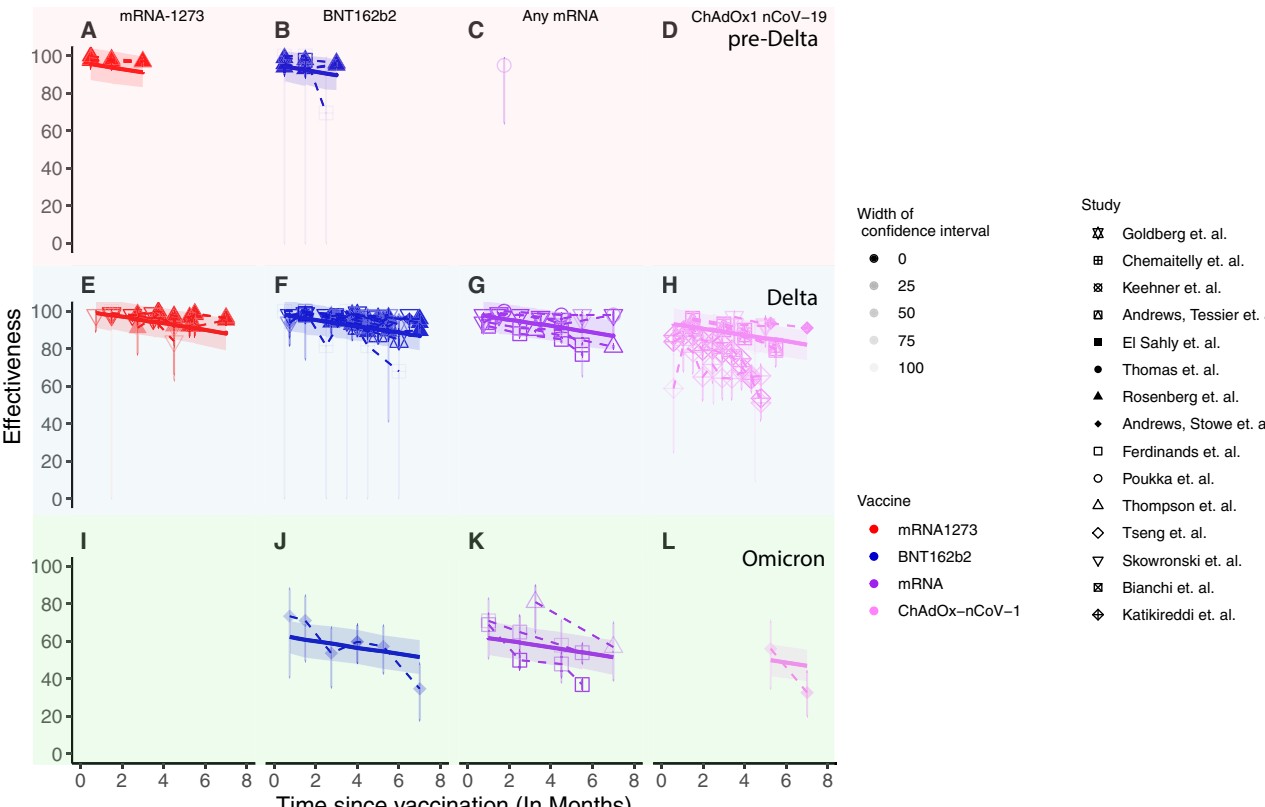

**Fig. 2 | Results of linear mixed effects regression model fit to vaccine effectiveness data against severe COVID-19 extracted from our systematic review.** Vaccine effectiveness data against severe COVID-19 (points and whiskers for 95% CI) are shown for pre-Delta (top row, panels (**A**)–(**D**)), Delta (middle row, panels (**E**)–(**H**)) and Omicron (bottom row, panels (**I**)–(**L**)) variants. Note that for panels (**D**) and (**I**), no effectiveness data was available. Opacity indicates the degree of confidence in the data as determined by the width of the confidence interval. Predicted vaccine effectiveness using a linear mixed effects regression model (solid lines) and 95% confidence intervals (shaded area) are overlaid. The figure shows effectiveness following mRNA-1273 (red), BNT162b2 (blue), any mRNA (purple) and ChAdOx-nCov-1 (pink) vaccination.

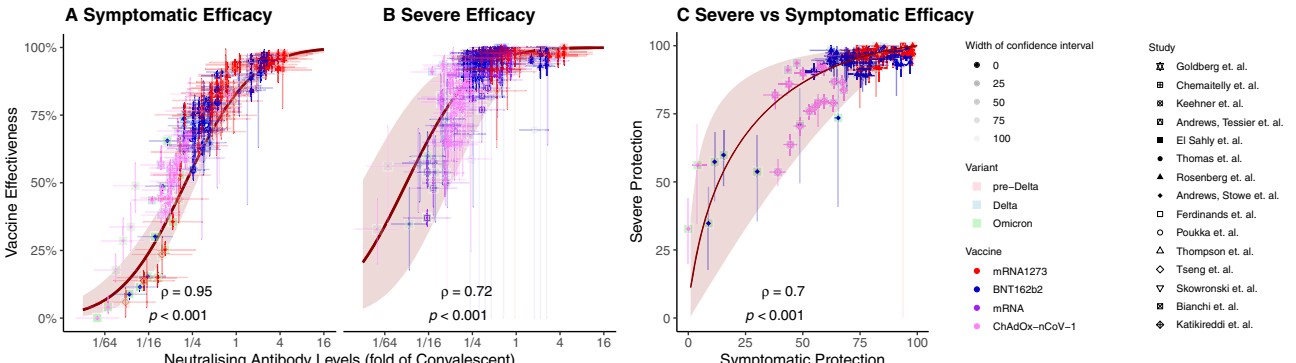

**Fig. 3 | Correlation between estimated neutralising antibody titres and vaccine effectiveness.** Correlation between estimated neutralising antibody titres (accounting for vaccine used, variant studied and time since vaccination) and clinical data for **A** vaccine effectiveness against symptomatic SARS-CoV-2 infection, **B** vaccine effectiveness against severe COVID-19. **C** Correlation between vaccine effectiveness against symptomatic and severe COVID-19. Solid lines indicate the predicted relationship taken from ref. [4], and shading indicates 95% CI of the model estimates. *X*-axis confidence intervals in (**A**) and (**B**) represent the degree of confidence in the estimate neutralising antibody titre. A breakdown of the relationship shown in panels (**A**) and (**B**) by variant and study type is shown in Supplementary Fig. S1. Figure shows effectiveness following mRNA-1273 (red), BNT162b2 (blue), any mRNA (purple) and ChAdOx-nCov-1 (pink) vaccination against pre-Delta (pink background) Delta (blue background) and Omicron (green background) variants.

measure efficacy in clinical studies. For all 17 of these data points, the reported efficacy against severe disease was above 93%. It is important to note that this analysis, unlike the analysis presented in the previous section, does not directly estimate the correlation between neutralising antibodies and protection. Rather, it tests the model's prediction of the relationship between protection from symptomatic infection and protection from severe infection (based on the relationship of each to neutralising antibody titre). These findings show that vaccine effectiveness against severe COVID-19 is not decoupled from effectiveness against symptomatic infection but is highly correlated and predictable based on the previously published relationship between neutralisation and protection[4].

## Discussion

Neutralising antibodies are an important immune correlate of vaccine-mediated protection from symptomatic SARS-CoV-2 infection[4–7,16]. However, their role as a correlate of protection from severe SARS-CoV-2 infection has been less clear. One challenge in determining whether neutralising antibody titres are associated with protection from severe COVID-19 is that the predicted 50% protective titre is below the limit of detection for many in vitro neutralisation assays[4,32,33]. However, by adjusting reported neutralising antibodies to incorporate the effects of immune waning and recognition of circulating variants and then correlating these with observed protection, we show that predicted neutralising antibody titres are strongly correlated with reported estimates of protection against severe COVID-19 disease (Spearman's $\rho = 0.72$, $p < 0.001$).

The initial work identifying neutralising antibodies as a correlate of protection from SARS-CoV-2 relied on analysis of data from the Phase 2 and Phase 3 trials of seven vaccines, of which two were mRNA vaccines, one was protein-based, three were viral-vector vaccines, and one was a whole virus vaccine[4]. This work found that while a level of neutralising antibodies equivalent to 20% of the GMT of early convalescent subjects (around 54 IU/ml) was associated with 50% protection from symptomatic infection, protection from severe infection was predicted to be achieved with a 6.5-fold lower titre (3.1% of convalescent, around 8 IU/ml)[4]. Unfortunately, the 50% protective level for symptomatic infection is close to the detection limit in most assays reported in phase 1/2 vaccine studies. Similarly, the 50% protective titre from protection against severe infection is below the limit of detection in 5/7 of the reported assays[4]. This low sensitivity of neutralisation assays arises largely because of the relatively high serum dilutions used in most in vitro assays, with a serum dilution of 1:10 or 1:20 being the lowest tested in most cases[34–36]. The relative insensitivity of the in vitro neutralisation assays has led to a perception of "protection in the absence of neutralisation"[37]. However, many subjects have clearly measurable antibody levels using antibody binding assays, even when these are not detectable by neutralisation assays[38]. This shows that an 'undetectable' in vitro neutralisation titre is reflective of the limit of detection of the assay and does not necessarily indicate the absence of neutralising antibodies. Thus, protection from severe SARS-CoV02 infection in the absence of detectable in vitro neutralisation, while perhaps not intuitive, is actually a clear prediction of the model[4].

Although the association between neutralising antibodies and protection from symptomatic SARS-CoV-2 infection has been investigated in several settings[4–7,14,16], protection from severe infection has heretofore remained more difficult to unravel. Our first analysis of the relationship between neutralising antibodies and protection from COVID-19 was parameterised based on Phase 2 (immunogenicity) and phase 3 (efficacy) data from seven vaccines and from convalescent individuals[4]. However, none of these studies reported a protective efficacy below 50% for symptomatic infection or below 85% for severe COVID-19. Thus, the model estimates of efficacy at low neutralising antibody titres were an extrapolation from the data available at the time. In addition, all phase 2 and phase 3 licensure studies reported responses and protection against the ancestral virus in the first months after vaccination. Here we aggregate the available epidemiological data to investigate whether the relationship between neutralising antibodies and protection from severe COVID-19 remains predictive across a diverse range of real-world scenarios of different vaccines, variants and time since immunisation. Our analysis demonstrates that reported changes in neutralising antibodies over time and against different variants are indeed predictive of changes in protection from severe COVID-19 across these different scenarios, at least for the vaccines captured in our systematic review (Fig. 3). It is notable that the previously developed model appears highly predictive of protection at low neutralising antibody levels, in the presence of waning immunity

and immune escape variants (Fig. 3A, B), providing an important validation of the model. This provides strong, though indirect, evidence that neutralising antibodies are a correlate of protection from severe COVID-19.

Observations that vaccine protection against severe COVID-19 is relatively maintained against variants (compared to protection from symptomatic infection), combined with the fact that protection against symptomatic infection appears to wane faster than protection against severe COVID-19[8,9,26] has led some to conclude that there is a 'decoupling' in protection from symptomatic and severe disease[13]. However, this appears built on an expectation that protection will wane in parallel for different COVID-19 severities. Here we show that protection against severe COVID-19 is in fact strongly coupled with and predictable from the protection against symptomatic infection (albeit in a nonlinear fashion) (Fig. 3C) and that this relationship is consistent with predictions of the previously published model of the relationship between neutralising antibody titres and protection (Fig. 3A, B)[4].

That antibodies are capable of reducing the risk of severe COVID-19 is perhaps not surprising since monoclonal antibody therapy administered in the first five days after symptom onset has been shown to reduce the risk of hospitalisation for severe SARS-CoV-2 by up to 85% when administered in doses comparable to the neutralisation titres achieved in individuals vaccinated with an mRNA vaccine and/or booster (i.e., 7-fold the neutralisation titre found in the average convalescent individual)[39,40]. Indeed, some studies suggest that passive antibody administration may remain effective even later in infection when administered to seronegative subjects[41]. Together this suggests that spike-specific antibodies may play a mechanistic role in protection from progression from symptomatic to severe COVID-19.

We and others have previously shown that there is a good correlation between neutralising antibodies and protection from symptomatic COVID-19[4,14,16]. In addition, we previously identified a similar relationship between neutralising antibodies and protection from severe COVID-19, although this was based on a small amount of data on severe infection[4]. Importantly, the lowest reported titres were around 20% of convalescent antibodies, and so predicted efficacies against severe disease for neutralising antibodies below this level are based on extrapolation only. In this study, we confirm that the relationship between neutralising antibodies and protection from severe COVID-19 is maintained as neutralising antibody titres change over time and against specific SARS-CoV-2 variants. Passive antibody studies have also directly demonstrated a mechanistic role for antibodies in protection from severe COVID-19. Administration of antibodies during symptomatic SARS-CoV-2 infection can reduce the risk of subsequent hospitalisation or death by up to 85%[40] (reviewed in ref. [42]). This demonstrates a direct role for antibodies in reducing infection severity, independent of their role in preventing the acquisition of infection. However, this does not prove that antibodies are exclusively responsible for protection against severe disease, and we cannot exclude the possibility that there are alternate mechanisms, such as T-cells, that also contribute to protection. Some evidence has suggested a potential role for T-cell responses when neutralising antibody responses are not detected[43]. However, since cellular responses and neutralising antibodies typically correlate, it is difficult to determine whether T-cells are causal in this instance or merely correlated with a level of neutralising antibodies that is below assay detection[44]. In addition, since T-cell help is required for the generation of high titre neutralising antibody responses, they likely play an important indirect role in protection. Therefore, while we can conclude that neutralising antibodies are associated with protection from severe disease (this study) and that passively administered antibodies can reduce severe diseases[42], more work is still required to determine the contribution of cellular immune responses to protection.

This study has a number of limitations. First, it aggregates the available epidemiological studies, which are heterogeneous with

respect to vaccines, variants and study methodology (see Table 1). In addition, these real-world observational studies report vaccine effectiveness (in a nonrandomised setting), whereas the previous relationship between neutralisation titre and protection was derived from analysis of vaccine efficacy in randomised controlled trials[4]. We have previously shown that observational test-negative case-control studies (TNCC) tend to report a higher level of protection than seen in randomised trials[16], although the effects of the different study designs used in our analysis are unclear (Supplementary Fig. S1). We also note that the studies identified in our systematic review all reported on either mRNA or viral-vector vaccines, and we did not identify any reports of efficacy over time after vaccinating with inactivated or protein-based vaccines. Some studies of vaccine effectiveness of inactivated vaccines against the omicron variant suggest that effectiveness may be higher than expected due to neutralising antibodies alone[45,46], although these studies did not meet the criteria for our systematic review as they were published after the date cut-off. Further analysis is required to determine whether the correlation between neutralising antibodies and protection against severe disease is different in some way for inactivated vaccines. Another limitation is that neutralising antibody titres were not directly measured in the study populations. Instead, neutralisation titres at different times and against different variants had to be estimated based on published parameters (see supplementary materials and Supplementary Tables S3 and S4). However, in the absence of a widely accepted standard assay for in vitro neutralisation, it is unlikely that, even if neutralisation titres had been measured, they would have been comparable between studies[33]. Future studies directly measuring titres in exposed populations may be needed to directly validate our observations. However, given the low frequency of severe SARS-CoV-2 infection and the fact that the neutralisation levels associated with protection from severe COVID-19 are below detection in many assays, such studies may prove challenging.

The recent Omicron BA.1 and BA.2 waves across much of the world have illustrated the capacity of the virus to evade humoral immunity at the population level, with many previously effective vaccines showing low levels of protection from symptomatic Omicron variant infection[8]. Our analyses included data from three studies reporting on vaccine effectiveness against severe COVID-19 from the Omicron variant. We found that neutralising antibodies remain predictive of protection under this scenario[8,22,23] (Supplementary Fig. S1, panels M and N, Spearman's correlation $p < 0.003$ for both symptomatic and severe COVID-19). The relative maintenance of protection from hospitalisation and death has greatly reduced the public health burden of infection[22]. It will be increasingly important to understand disease severity as an endpoint in future epidemiological monitoring studies. In this context, identifying neutralising antibodies as a robust correlate of protection against severe COVID-19 is both timely and essential. In conclusion, we show that the relationship between neutralising antibody titres and protection holds across the spectrum of COVID-19 severity and that neutralising antibody titres are predictive of protection against symptomatic as well as severe COVID-19.

## Methods
### Mixed effects model fitting
To determine if vaccine effectiveness against severe COVID-19 was dependent on vaccine, variant and/or time since vaccination, we fit a mixed effects model to vaccine effectiveness with vaccine and variant as categorical covariates, time as a continuous covariate and included a random effect for the study from which the data came. The model was:

$$Eff = Eff_{base} - A_i - B_j - C_j t + \zeta_{study} \qquad (1)$$

where $A_i$ is a vaccine-specific adjustment for vaccine $i$, $B_j$ is a variant-specific adjustment for variant $j$, $C_j$ is a variant-specific parameter determining the change in effectiveness over time since vaccination ($t$) and $\zeta_{study}$ is a random effect for the study from which the data came. Values of these parameters are given in Supplementary Table S1.

### Estimating mean neutralising antibody titres
We estimated the mean neutralising antibody titre that would be associated with each real-world effectiveness data point. This estimated neutralising antibody titre was based on:

1. The vaccine that was administered
2. The variant against which effectiveness is being measured
3. The time since vaccination
4. The dosing schedule for the vaccine
5. The timeframe over which efficacy was reported in the original phase 3 trials compared to the timeframe measured in the extracted real-world data points.

We then combined these factors into an estimate for the mean neutralising antibody titres that would have been observed over the time period that matches the reported effectiveness. Detailed equations describing how these factors were used to estimate neutralising antibodies are given in the supplementary materials.

### Determining confidence intervals using parametric bootstrapping
Confidence intervals of all estimates for neutralising antibody titres and predicted efficacies (shaded regions) in Figs. 2, 3, Supplementary Figs. S1–S4 were generated using parametric bootstrapping on the parameters with uncertainty in their estimation (as previously reported in ref. [16], detailed in Supplementary Methods using parameters in Supplementary Tables S3 and S4).

### Statistical analysis
All statistical comparisons were performed using R (version 4.0.2). Tests performed were Spearman's rank correlations unless otherwise stated.

### Reporting summary
Further information on research design is available in the Nature Portfolio Reporting Summary linked to this article.

## Data availability
Data used in this analysis is available at https://github.com/InfectionAnalytics/Predicting-Effectiveness-Against-Severe-COVID19.

## Code availability
Code used in this analysis is available at https://github.com/InfectionAnalytics/Predicting-Effectiveness-Against-Severe-COVID19.

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

## Acknowledgements

This work would not be possible without the many scientists who generously provided the published data analysed in this study by making the

data directly available through the original publication. The authors thank these scientists for their contribution, and the individual sources of data are indicated in the references and supplementary tables. This work was supported by Australian NHMRC program grant 1149990 to S.J.K. and M.P.D., an Australian MRFF award 2005544 to S.J.K. and M.P.D., and MRFF 2015313 to S.C.S. and M.P.D. S.J.K., D.C. and M.P.D. are supported by NHMRC Investigator grants. D.S.K. is supported by a University of New South Wales fellowship.

## Author contributions

D.C., M.P.D. and D.S.K. contributed to the study design. D.C. and S.R.K. designed and performed the systematic review. D.C. and M.S. performed data extraction and curation. D.C., A.R., D.S.K. and T.E.S. performed the data analysis. D.C., M.P.D., D.S.K., S.J.K. and S.C.S. contributed to shaping the direction of the work. All authors contributed to the writing and reviewed and approved the final report.

## Competing interests

The authors declare no competing interests.

## Ethics

This work was approved under the UNSW Sydney Human Research Ethics Committee (approval HC200242).
