## [Peer Review File · Nature Communications]

Predicting vaccine effectiveness against severe COVID-19 over time and against variants: a meta-analysisREVIEWER COMMENTS

Reviewer #1 (Remarks to the Author):

The authors extended their previous work published in Nature Medicine which used neutralising antibodies (nAbs) to predict vaccine effectiveness (VE) against severe COVID-19 over time and against variants. The authors aimed to predict VE against severe COVID-19, the high correlation was however partially concluded from the predictions of nAbs and VE against symptomatic infections ($r = 0.93$).

The research question that the authors trying to answer is important, while the methods used to approach the answer are subject to several limitations and are against existing evidence, which could affect the reliability of the findings and conclusions. The proposed framework made significant contributions to aid us understand the vaccine included nAbs and VE of symptomatic cases early in the pandemic, while extrapolating it to severe infections may need more careful and comprehensive considerations.

Despite that the nature publishing group encourages for open data and that the data used in this work were largely from literature without obvious ethical concerns, it is hard to replicate the analyses based on current descriptions in methods and underlying data provided.

Major comments:

- 1) The extrapolation of their previous model predictions ignored and against the exiting biological evidence that T cell immunity levels correlates with severe outcomes, and that neutralising antibodies were poorly correlated with severe outcomes. Extreme examples are that naive people who were seronegative but had pre-existing cross-reactive T cell immunity (from seasonal coronaviruses) were less likely to develop into severe diseases. These evidence suggested that T cell immunity is played a critical role in preventing severe outcomes, which was failed to be accounted by the model.
- 2) The authors stated that a meta-analysis was performed to extract real-word VE for model validation. I am concerned that the fact that the meta-analysis was carried out in a non-systematic way and lack of clear criteria may lead to selection biases. Why inactivated vaccines were not included was not justified. There are several publications appeared to meet the inclusion criteria, but were not included in the analyses (e.g., Collie 2022 NEJM; Ranzani OT 2022 medRxiv; Ranzani OT 2021 BMJ; McMenamin ME 2022 Lancet Inf Dis).
- 3) The majority of the data points used in model predictions were from the higher end of VE, which may bias the correlation that reported in this study. The VE at low nAbs level may suffer from edge effects in the original Nat Med model, therefore predictions in that area may be more challenging. Such area is where nAbs induced by inactive vaccines fell, which was excluded by the authors. In fact, the previously published model would predict a 20% to 30% VE against severe outcomes for two doses inactive vaccines. However, a number of real-word studies suggested that two doses inactive vaccines, despite or the undetectable cross-reactions against Omicron strains, would provided VE as high as 90% (McMenamin ME 2022 Lancet Inf Dis).
- 4) Multiple data points from a single study (despite that the age group or time since vaccination may differ) seemed to be used to derive the reported correlations. It is not clear whether the authors adjusted from clustering of studies. If not, the high correlation may be biased by few studies that disproportionally contributed data points.
- 5) It is not clear at what scale the correlation between predicted nAbs and VEs were calculated. Since most of the data points shown in Figure 3 clustered in the high end, whether the correlation was calculated in log or linear scale may make difference. This needs to be described and justified.

Minor comments

- 1) It is not clear how many vaccine doses were used in this study. Did the authors also consider boosters?
- 2) For data extracted from meta-analysis (especially TND), how did the authors deal with

population natural infection histories in the study populations?

3) Line 86, need to explicitly describe the searching, screening and inclusion process. 15 studies seemed not representative for current literatures.

4) Line 201 & 208 Correlation value should also be reported.

5) Line 226-228 It's unclear how this analyses was performed. Isn't the correlates model originally fitted based on nAbs?

6) Line 239 Correlation is for symptomatic and severe diseases combined? This study seemed to focus on V_e against severe COVID-19, so not sure why symptomatic was also included.

7) Lines 245-248. There are clear opposite real-world evidence suggested very high VEs against severe COVID-19 after one or two doses inactivated vaccines.

Reviewer #2 (Remarks to the Author):

This excellent paper extends earlier, highly influential publications from the research group on the role of humoral immunity in protection against SARS-CoV-2 infection and disease. The basis of the present paper is the use of published methods to further analyze the protective role of virus neutralizing antibodies (NABs). A previous report established the normalized NAb titer as the dominant Correlate of Protection (CoP) against SARS-CoV-2 infection in vaccine trials. That finding was not only important, it made great sense based on long standing knowledge of how NABs act. Now, the authors show that NABs also protect against severe Covid-19, and at lower titers than are required to protect against infection. That too is an important finding, and it is also one that is consistent with fundamental knowledge of the immune response to viral pathogens. Additional analyses address the also key topic of virus variants with various degrees of NAb resistance; and the effect of the waning of NAb titers over time. Overall, this paper is impressive in respect of its scope, its significance and its fundamental soundness.

I have no suggestions about how to modify the text.

The authors are, I assume, well aware that researchers studying T-cell immunity have a tendency to talk up what they work on. They often make the point that T-cells protect against viral disease, including Covid-19. However, my own read of the Covid-19 vaccine literature is that the hard evidence supporting these claims is thin - almost to the point of non-existence. The authors' work in this area is of particular importance in that context. If I may make so bold as to suggest the next area of analysis for the present research team, it is to look at the ever-increasing amount of T-cell response data available from the same vaccine trials they analyzed in the present paper. Is there a way to apply the same techniques? My prediction is that they would find scant evidence for T-cell responses being a CoP for Covid-19 vaccines, either in infection (no surprise) or disease (an eye-opener for some researchers).

Reviewer #3 (Remarks to the Author):

The authors used data from previously published studies on SARS-CoV-2 vaccine effectiveness to firstly predict levels of neutralizing antibodies over time. They then demonstrate an association between these predicted levels of neutralizing antibodies and vaccine effectiveness against both symptomatic and severe outcomes. This an interesting study, and the authors have a good rational for why they are using modeling and combining data from different studies to determine if there is an association between neutralizing antibodies and vaccine effectiveness against severe covid-19. They use a number of different modeling techniques to test the robustness of their results with consistent findings. The conclusions are supported by the data presented and the main limitations are discussed. However, in parts of the results sections it was challenging to follow exactly which modeling

strategy was being used without referring the supplementary material. The authors report in the results section that they identified and extracted data on 15 studies, but there are only very sparse details either here or in methods section on how these studies were actually found, what literature did they search? What time period did they use? Which search terms? What studies were excluded & why? Did they perform a systematic review or how were the studies selected?

There is a footnote in the Supplementary that “†Denotes studies that included efficacy against confirmed infection without reporting on symptoms. These data were not included in the final analysis.” It appears next to 6 of the studies – so were only 9 studies included in the analysis? It is not clear which parts of the analysis these studies were excluded from.

Figure 2 and Figures 3 for example reading the legend it appears to show data from 12 of the studies (with Skowronski split in two)?

A flow chart and perhaps referring to the PRISMA guidelines may help with this

For the first part of the paper, neither the introduction nor methods clearly explain why vaccine regimen, time since vaccination and variant are the key parameters of interest or whether there were any other variables considered that did not make it into the final model/ study selection?

Minor comments:

Title: The title should be revised to state what type of study it is

Introduction line 51: The authors state: ‘ Studies of the relationship between neutralising antibodies and protection from symptomatic SARS-CoV-2 infection...’ but they only reference one of their own papers in this sentence. Suggest to be rephrased to ‘we have previously...’ or add additional references for clarity

Introduction line 80: Suggest the sentence is revised to outline the study aims and hypothesis rather than stating the results (these statements should be reserved from the results and discussion).

Consider moving table S1 to the main manuscript, could move Figure1 to the supplementary instead

Results line116, the authors state that vaccine effectiveness was lower with the Delta variant compared to pre-Delta however, the confidence intervals overlap 1.

Figure 2. There is a typo in the footnote

Reviewer #4 (Remarks to the Author):

This is a very important and innovative paper that uses available data to examine the relationship between covid-19 vaccine type and timing protection against severe disease from different variants. Then, the authors use a model of neutralizing antibody titers to predict the relationship with variant-specific protection against severe outcomes. It is a logically sound approach, but, as the authors acknowledge, it combines two relationships - first that efficacy wanes over time in a variant- and vaccine-specific manner, that neutralizing antibodies are a predictor of vaccine effectiveness and that neutralizing antibodies wane over time. Then, they essentially combine these relationships and assert that neutralizing antibodies can predict efficacy. Generally, it is a well-performed analysis that fills a knowledge gap, but does require a strong assumption of a causal relationship of vaccine -> neuts -> efficacy/effectiveness.

I do have some specific concerns about how the studies were selected and analyzed.

1. This is a systematic review but the literature search is not presented in sufficient detail to understand the inclusion and exclusion criteria and how the search strategy was implemented. I strongly suggest that the PRISMA guidelines be followed.
2. The traditional checks that are performed in a meta analysis were not done or not reported. A major question is whether there is too much heterogeneity between the studies for them to be combined. The substantial number of outliers suggest that may be the case. I suggest a formal analysis of I^2 or other such metrics.

Minor comment:

Fig 3. The different symbols for each study type are not helpful since they are so small and largely clustered in one region of the graph.

REVIEWER COMMENTS

Reviewer #1 (Remarks to the Author):

The authors extended their previous work published in Nature Medicine which used neutralising antibodies (nAbs) to predict vaccine effectiveness (VE) against severe COVID-19 over time and against variants. The authors aimed to predict VE against severe COVID-19, the high correlation was however partially concluded from the predictions of nAbs and VE against symptomatic infections ($r = 0.93$).

The research question that the authors trying to answer is important, while the methods used to approach the answer are subject to several limitations and are against existing evidence, which could affect the reliability of the findings and conclusions. The proposed framework made significant contributions to aid us understand the vaccine included nAbs and VE of symptomatic cases early in the pandemic, while extrapolating it to severe infections may need more careful and comprehensive considerations.

Despite that the nature publishing group encourages for open data and that the data used in this work were largely from literature without obvious ethical concerns, it is hard to replicate the analyses based on current descriptions in methods and underlying data provided.

All the data and code were made available to the reviewers so that a reader can exactly replicate our analysis using the original data and code, however we appreciate that it may have been difficult to determine exactly which part of the code related to which calculations and plots presented in the manuscript. We have therefore now expanded upon the 'read me' file and more clearly clarified the precise location in our code where calculations are made, and have also added extra commenting to our code that further made it clear where each calculation presented in the manuscript was derived from the code.

In addition, the large amount of data may have made it difficult for a reader to visualise the contribution of individual studies to the overall analysis in the main figure (for readers who may not wish to use the R-code). Therefore, we have also added additional supplementary figures (figure S5-8), which present the data within each study more clearly.

Major comments:

1) The extrapolation of their previous model predictions ignored and against the exiting biological evidence that T cell immunity levels correlates with severe outcomes, and that neutralising antibodies were poorly correlated with severe outcomes. Extreme examples are that naive people who were seronegative but had pre-existing cross-reactive T cell immunity (from seasonal coronaviruses) were less likely to develop into severe diseases. These evidence suggested that T cell immunity is played a critical role in preventing severe outcomes, which was failed to be accounted by the model.

We agree that several lines of evidence point to a potential role for T cells in protection from disease and we now make this point more clearly in our revised manuscript. An even more comprehensive model of immunity from severe infection incorporating multiple aspects of

immunity to severe infection would be most welcome. However, the available data to date do not easily allow themselves to be incorporated into a model yet since (a) measurements of T-cell immunity are difficult to standardise across studies (we note this was challenging even for neutralising antibody assays, and relied on a ‘standardisation’ based on neutralising antibodies in convalescent subjects) (b) more limited data sets on T cell immunity are available from vaccine studies (c) other lines of evidence such as pre-existing cross reactive T cells are rarely studied. We have recently written on this topic, noting that although it is possible T cells are important for protection against severe outcomes, there is a current absence of clinical evidence for a role of T cells in protection from symptomatic or severe COVID-19 (e.g. Kent et. al. *Nat Rev Imm* and references therein). We also differ from the reviewer on some points raised.

For example, the reviewer states that our analysis is “*against the exiting[sic] biological evidence that T cell immunity levels correlates with severe outcomes*” (but provides no citation for evidence of this). In addition, the reviewer states that “*neutralising antibodies are poorly correlated with severe outcomes*”. However, as far as we are aware the study we are reporting here is the only analysis that attempts to systematically address this very question, indicating its importance in the literature. In addition, the reviewer also refers to the role of “*pre-existing cross-reactive T cell immunity (from seasonal coronaviruses)*.” However, this study of infection in (immunologically) naïve individuals does not directly relate to our study vaccine-mediated protection from COVID-19.

The question about whether protection against severe COVID-19 disease is mediated by antibodies, T-cells or a combination of the two, is indeed a very controversial question, and one for which there is currently a lack of definitive evidence either way. We feel that the work in this manuscript contributes importantly to this discussion – although we agree it does not definitively resolve the question. Instead, we tested, for the first time, whether the loss of neutralising antibodies over time and against different variants, after primary vaccination, does or does not correlate with the loss of vaccine effectiveness. We find that the changes in neutralising antibodies are indeed correlated with protection from severe outcomes. This is the first test (that we are aware of) of whether neutralising antibodies are a good or poor predictor of protection from severe outcomes. The data and analysis we have accumulated suggest they are a good predictor (though we do not claim this implies they are mechanistic).

We note that the comment from Reviewer 1 is in direct contrast to the comment from Reviewer 2 below, who noted that “*researchers studying T-cell immunity have a tendency to talk up what they work on. They often make the point that T-cells protect against viral disease, including Covid-19. However, my own read of the Covid-19 vaccine literature is that the hard evidence supporting these claims is thin – almost to the point of non-existence.*” These two very divergent views on the role of T-cells are indeed representative of the views of the scientific community at large (and the strong feelings of many on this topic), and therefore it is indeed important to address this question, which we have now done in the discussion.

We thank the reviewer for bringing to our attention the fact that it would be a good idea to specifically comment on the evidence for and against a T-cell contribution to protection against severe disease.

The discussion now reads:

We and others have previously shown that there is a good correlation between neutralising antibodies and protection from symptomatic COVID-19^{4,14,16}. In addition, we previously identified a similar relationship between neutralising antibodies and protection from severe COVID-19, although this was based upon a small amount of data on severe infection⁴. In this study, we confirm that the relationship between neutralising antibodies and protection from severe COVID-19 is maintained as neutralising antibody titres change over time and against specific SARS-CoV-2 variants. Passive antibody studies also suggest a mechanistic role for antibodies in protection, since administration of antibodies alone can reduce severe infection^(39 and references therein). However, this does not prove that antibodies are exclusively responsible for protection against severe disease, and we cannot exclude the possibility that there are alternate mechanisms, such as T-cells, that also contribute to protection. Some evidence has suggested a potential role for T cell responses when neutralising antibody responses are not detected⁴⁰ However, since cellular responses and neutralising antibodies typically correlate it is difficult to determine whether T cells are causal in this instance or merely correlated with a level of neutralising antibodies that is below assay detection⁴¹. Therefore, while we can conclude that neutralising antibodies are associated with protection from severe disease (this study), and that passively administered antibodies can reduce severe diseases³⁹, more work is still required to determine the contribution of cellular immune responses to protection.

2) The authors stated that a meta-analysis was performed to extract real-world VE for model validation. I am concerned that the fact that the meta-analysis was carried out in a non-systematic way and lack of clear criteria may lead to selection biases. Why inactivated vaccines were not included was not justified. There are several publications appeared to meet the inclusion criteria, but were not included in the analyses (e.g., Collie 2022 NEJM; Ranzani OT 2022 medRxiv; Ranzani OT 2021 BMJ; McMenamin ME 2022 Lancet Inf Dis).

We thank the reviewer for requesting an improved presentation of our literature search – we feel our work has been much improved by bringing our search in line with the PRISMA guidelines for a systematic review, as described above.

In addition, we note that the reviewer is suggesting that our results are perhaps dependent on the studies we have chosen to include and exclude in our meta-analysis, and the potential for selection bias. We apologise that we did not previously present our work in sufficient detail to make the systematic nature of our approach clear. We have now addressed this concern in two ways. We have now

1) Repeated our systematic search following PRISMA Guidelines and present these results in a clearer format.

As with a formal PRISMA-defined systematic review, the aim of our literature search has always been to exhaustively capture all eligible studies of effectiveness which met our inclusion criteria of reporting on the decay of effectiveness against a defined variant, in order to avoid selection bias. However, we had previously presented insufficient detail of our methods of performing an exhaustive search and had not shown that it was exhaustive.

As per the suggestions of both Reviewer 1 and Reviewers 3 and 4 below we have repeated our search of the literature using the same search terms and over the same time interval as our original search, while following the PRISMA guidelines for a systematic review.

Specifically, as per the PRISMA guidelines for a systematic review, we have:

- 1) Specified the inclusion and exclusion criteria studies
- 2) Specified the databases and other sources searched or consulted to identify studies.
- 3) Specified the date up to which the search was conducted.
- 4) Presented the full search strategy used
- 5) Specified the methods used to collect data from the reports
- 6) Listed outcomes for which data were sought.
- 7) Listed other variables for which data were sought.
- 8) Described the results of the search and selection process, including the number of records identified in the search and the number of studies included in the review, and have included a flow diagram (as Figure S9 and reproduced below)

These improvements of our meta-analysis search strategy identified two new studies and better ensure an exhaustive ascertainment of relevant studies, in line with the PRISMA guidelines. The methods are fully described in the Supplementary materials, and the details of our literature search are shown in the appropriate PRISMA flow diagram in Figure S9 (figure shown above).

The fact that in this repeated analysis we only identified an additional two papers that met the criteria for inclusion shows our original search was nearly exhaustive in capturing the literature

on this topic at the time. The additional two papers have been added to the analysis and do not change the results and indeed reinforce the robustness of the initial conclusions.

The updated supplementary methods describing the systematic review now reads:

Literature search for data on vaccine effectiveness

To be included in our analysis, studies must have included data on the efficacy or effectiveness against a defined symptomatic COVID-19 and/or severe COVID-19 clinical endpoint, of a primary COVID-19 vaccine schedule in humans, compared to an unvaccinated control population and over time (i.e. present a time series of efficacy/effectiveness). In addition, these studies must have efficacy/effectiveness data reported:

- (i) for a single vaccine (or vaccine type, e.g. mRNA vaccines) or for multiple vaccines with data stratified by vaccine,*
- (ii) for an identifiable variant that could be identified as either occurring entirely before the delta wave, or during one of the delta or omicron wave, or with data stratified by variant wave,*
- (iii) with an identified time since vaccination, and*
- (iv) be included explicitly in the publication, or be in or readily extractable from the original publication.*

Studies (or data within a study) were excluded if they:

- (i) Did not present a primary report of vaccine efficacy or effectiveness (e.g. secondary analysis of other studies were excluded as well as review articles, perspectives, opinions)*
- (ii) Did not report vaccine efficacy or effectiveness compared with an unvaccinated control population (e.g. studies that compared vaccine efficacy/effectiveness of one vaccine against another vaccine)*
- (iii) Did not report vaccine efficacy or effectiveness over multiple time points since vaccination*
- (iv) Reported vaccine efficacy/effectiveness for a mixture of vaccines of different types/platforms*
- (v) Reported vaccine efficacy/effectiveness against a mixture of, or with unspecified, circulating SARS-CoV-2 variants (with the exception of variants occurring before the delta wave, which were allowed to vary).*
- (vi) Reported vaccine efficacy/effectiveness only for individuals deemed at a high risk of COVID-19 (including studies of exclusively individuals >80 years of age)*

We identified 488 potential non-duplicate studies, of which 376 were excluded after screening titles. A total of 112 records were retrieved and assessed. Of these, 96 were subsequently excluded (reasons identified in Figure S9), with a total of 15¹⁻¹⁵ eligible studies identified and included in our meta-analysis. The complete flow diagram showing the review process is outlined in Figure S9.

Extraction of vaccine effectiveness data

For each identified study we recorded any reported measure of vaccine effectiveness against symptomatic or severe COVID-19 disease. We also recorded the time post vaccination at which these estimates were derived, the vaccine used, the variant against which effectiveness was measured, the age of participants, the type of study and the country in which the study was conducted.

Data was extracted from the eligible studies independently by two of the study co-authors (DC and MS). Data was extracted from identified papers either directly from tables included within the publication, or else was extracted from figures using WebPlotDigitizer¹⁷ The location within the original publication of the data used for each study is indicated in Table 1. We did not include effectiveness data taken from cohorts classified as at high risk for COVID-19 disease, or from cohorts exclusively comprised of individuals over 80 years of age, as it was felt that these cohorts may have different neutralising antibody responses, and there were not enough cohorts of such a nature to determine this definitively. We classified variants into pre-Delta (predominantly ancestral and Alpha), Delta and Omicron variants. We did not include data reporting on effectiveness exclusively against the Beta variant in the pre-Delta group, as neutralising antibody titres against the Beta variant have previously been identified to be vastly different to those against other pre-Delta variants, and we identified only one study reporting on effectiveness against the Beta variant, meaning that a separate analysis was not possible.

The 15 identified studies that met the above criteria, collectively provided 363 individual data points on vaccine effectiveness. The studies included effectiveness for three of the main vaccines used in primary vaccination regimes – mRNA-1273, BNT162b2 and ChAdOx1-nCoV-19, and on efficacy against symptomatic and severe COVID-19 disease outcomes. These studies are detailed in Table 1.

2) Tested the extent to which the studies referred to by the reviewer (that did not meet our search criteria) impact our conclusions.

The reviewer has suggested four papers that to include in our analysis, however, these studies did not fall within our defined criteria. The reasons these studies were not captured in our review are that two of them did not include a time course of vaccine effectiveness over time since second dose of the vaccine (Ranzani OT 2021 BMJ; McMenamin ME 2022 Lancet Inf Dis) and the other two were both published after the cut-off for our review (3 March 2022) (Collie 2022 NEJM; Ranzani OT 2022 medRxiv). Even though the papers suggested by this reviewer did not fall within the remit of our review, we here show that they are consistent with our results.

Although we cannot add these to our analysis (since they did not fall within our literature search, and including them would add a selection bias to our analysis), we did stress test our conclusions by adding the two studies suggested by the reviewer where a time course of vaccine effectiveness was available (Collie 2022 NEJM; Ranzani OT 2022 medRxiv). The remaining two studies (Ranzani OT 2021 BMJ , McMenamin ME 2022 Lancet Inf Dis) could not be included as they did not include a time course of effectiveness measurements

(presenting only an overall effectiveness estimate). In addition, in the case of McMenamin et. al., the study did not identify the particular variants circulating (aggregating data over multiple variant waves).

We show the results of adding data from Collie et. al. and Ranzani et. al. 2022 to our analysis in the figure below, which corresponds to figure 3 of the manuscript. In the figure below we have included all data that meets criteria for inclusion in our analysis (ie: the data in Figure 3 of the manuscript) in grey. We have added the data from the two additional studies proposed by the reviewer in colour. Panels A and B show the correlation between predicted neutralisation titre and reported effectiveness.

Even with these selectively identified studies included in our analysis, we find neutralisation still correlates with protection against severe outcomes and the relationships are still highly consistent with the model – with 9 of the 11 new points in the severe correlation having overlapping confidence intervals with the model (panel B of figure below).

We should note however, that for the Ranzani et. al. 2022 paper, it is difficult to predict the correct neutralisation titre after CoronaVac immunisation (and hence where the points should be placed along the x-axis). That is because our original model (Khoury et. al. *Nature Medicine*) was based on neutralising antibody titres from two-dose CoronaVac vaccination with a 14-day spacing interval between doses. However, the dosing interval used in the ‘real world’ population was generally longer (currently recommended as 8 weeks in Singapore, due to the low efficacy of the 2-week spacing), resulting in higher neutralising antibody titres and higher protection. In order to accurately represent these data (on the correct position on the x-axis relative to the other points), we would need additional information on the actual dosing intervals used in the study, as well as the neutralisation titres associated with this spacing (as we had for ChAdOx-nCoV-1).

However, we note that within our manuscript we also use an approach that is independent of the neutralisation titre, and just predicts the relationship between observed effectiveness against symptomatic and severe infection (Figure 3C of the main manuscript). Therefore, we also reproduced an equivalent version of panel C from Figure 3 of the main manuscript, as this offers a neutralising antibody independent way to compare efficacy against symptomatic and severe disease, and relate them to our original correlates model. We see in panel C above that for the data from Ranzani et. al. 2022, the relationship between protection from symptomatic disease and protection from severe disease is well aligned with the predictions from our correlates

model, with the points spaced either side of the predicted line, and 5/6 of the points lying within the 95% confidence intervals of our model. This confirms that even if the additional references chosen by the reviewer as counter-examples had conformed to our inclusion criteria, they would not have changed the conclusions of the analysis, and there is no evidence that protection against severe infection is higher than one might expect from protection against symptomatic infection.

Given the above two points, we are confident that our systematic search of the literature has not induced any unintentional selection bias into our analysis and results.

However, given the Reviewer's concerns about inactivated vaccines, we have also included a comment on this in the discussion:

We also note that the studies identified all reported on either mRNA or viral-vector vaccines, and we did not identify any reports of efficacy over time after vaccinating with inactivated vaccines. It would indeed be interesting to confirm whether this relationship is maintained for inactivated vaccines.

3) The majority of the data points used in model predictions were from the higher end of VE, which may bias the correlation that reported in this study. The VE at low nAbs level may suffer from edge effects in the original Nat Med model, therefore predictions in that area may be more challenging. Such area is where nAbs induced by inactive vaccines fell, which was excluded by the authors. In fact, the previously published model would predict a 20% to 30% VE against severe outcomes for two doses inactive vaccines. However, a number of real-world studies suggested that two doses inactive vaccines, despite or the undetectable cross-reactions against Omicron strains, would provided VE as high as 90% (McMenamin ME 2022 Lancet Inf Dis).

The author is correct that the original model (Khoury et al, *Nature Med*, 2021) was parameterised using data on vaccine protection against ancestral SARS-CoV-2 within the first 2-3 months of infection, and thus predominantly used data at the higher end of the efficacy / neutralising antibody spectrum. However, Figures 3A and B of the manuscript show that even in the lower efficacy / lower neutralising antibody range (bottom left of the figures) there is still good agreement between the data points and the model (and indeed the model has been tested on multiple occasions against lower efficacy data, for lower levels of neutralising antibodies (e.g. for variants of concern – Cromer et. al. *Lancet Microbe*, Cele et. al. *Nature*, Khoury et. al. *medRxiv*, 2021) and shown to remain accurate at these lower levels.)

It is also important to clarify here that in this manuscript we 'stress test' the original model by seeing if it remains predictive over different times since vaccination and against different variants (ie: at the lower end of the efficacy curve). At no point in this manuscript do we actually make any *predictions* from the efficacy data we have included in the model. Instead, we aim to test whether our previous predictions of the relationship between neutralisation and protection (Khoury et al, parameterised from data on ancestral virus) remains predictive as antibody titres wane over time and against different SARS-CoV-2 variants. The current work shows that efficacy estimates extracted from more recent studies are in good agreement with our predictions

made nearly 18 months ago (originally published in MedRxiv on 11 March 2021, and in *Nature Medicine* on 17 May 2021).

Finally, the reviewer states that our “previously published model would predict a 20% to 30% VE against severe outcomes for two doses inactive vaccines” however this is not correct. In fact, as can be seen in the figure above showing the Ranzani et. al. 2022 data, predicted efficacies against severe disease for inactivated vaccines much greater than this. In addition, in response to a later comment by this reviewer, we considered the effectiveness data presented for CoronaVac in the McMenamin study cited by the reviewer, and found that it is highly consistent with the model predictions (see response to point 7 below). However, we also think it is important to realise that the results of any individual study have specific nuances, and the strength of our approach is to aggregate data across multiple studies in as objective a way as possible.

4) Multiple data points from a single study (despite that the age group or time since vaccination may differ) seemed to be used to derive the reported correlations. It is not clear whether the authors adjusted from clustering of studies. If not, the high correlation may be biased by few studies that disproportionately contributed data points.

The reviewer is correct that, where appropriate, multiple data points from individual studies were used in this analysis. We think that the large amount of data in the main figures may have made it difficult to identify the data from different studies and led to concerns that patterns observed may not be seen in individual studies. To address this, we have now added supplementary figures (Figures S5-S8) that show the relationship between neutralisation and protection for individual studies for both symptomatic and severe disease.

Looking per study, we note that in the 20 panels of Figures S5-S8 for which data was available, we find that all panels with more than 6 contributing data points have positive correlations between predicted neutralising antibodies and vaccine effectiveness, and in all but one of these, this correlation is significant at the 95% confidence interval (and this one study has only 9 contributing data points).

In addition, in order to address the potential differential impact of individual studies, we have now also used a mixed effects model for our regression analysis that includes a random effect to account for study from which the data was derived. The inclusion of a random effect for study does not change the conclusions, namely that protection against severe disease is dependent on the vaccine used, the variant encountered, and the time since vaccination. Both the manuscript and the supplementary methods have been updated to reflect this.

The relevant section in the methods of the main manuscript now reads:

Mixed Effects Model Fitting

To determine if vaccine effectiveness against severe COVID-19 was dependant on vaccine, variant and/or time since vaccination we fit a mixed effects model to vaccine effectiveness with vaccine and variant as categorical covariates, time as a continuous covariate and included a random effect for the study from which the data came. The model was:

$$Eff = Eff_{base} - A_i - B_j - C_j t + \zeta_{study} \quad \text{Equation 1}$$

Where A_i is a vaccine specific adjustment for vaccine i , B_j is a variant specific adjustment for variant j , C_j is a variant specific parameter determining the change in effectiveness over time since vaccination (t) and ζ_{study} is a random effect for the study from which the data came. Values of these parameters are given in Table S1.

The relevant section in the results of the main manuscript now reads:

In a first analysis of the aggregated data, we used a linear mixed effects model to investigate the impact of vaccine type, variant, time since vaccination on vaccine effectiveness against severe COVID-19 (equation 1, Figure 2 and Table S1), while accounting for the potential random variation induced by using data from different studies. This showed that the reported effectiveness against severe COVID-19 did indeed vary by vaccine, variant and over time. For example, vaccination with mRNA-1273 showed higher effectiveness than vaccination with ChAdOx1 nCoV-19 (6.2% 95%CI 3.8 - 8.6). Similarly, effectiveness against severe COVID-19 was lower against Omicron than against either Delta or the pre-Delta variants (31.4% lower than against the pre-Delta variants, 95%CI 27 - 35.8). We also found that effectiveness declined over time since vaccination, with a decrease in effectiveness of 1.7% (95%CI 1.4 - 2.1%) per month.

5) It is not clear at what scale the correlation between predicted nAbs and VEs were calculated. Since most of the data points shown in Figure 3 clustered in the high end, whether the correlation was calculated in log or linear scale may make difference. This needs to be described and justified.

The correlations discussed in this manuscript between nAbs and VEs were Spearman correlations (i.e. non-parametric correlations, based on ranking), and therefore the correlation values are entirely independent of scale of the data (and whether it is on a log or linear scale). Spearman correlations are the most appropriate for this very reason and ensure the results are robust to the scale and clustering of the points. Pearson correlations were only used to assess the correlation between predicted and reported efficacies – a parametric comparison is appropriate in that case, because the expected and observed results can be measured on the same scale.

Minor comments

1) It is not clear how many vaccine doses were used in this study. Did the authors also consider boosters?

Within this manuscript we only considered the impact of two dose primary vaccination schedules – we have now made this clear in the text and in the supplementary methods. A major reason for this is that booster vaccination schedules become increasingly more complex than initial regimens, since the mixing of different vaccines and variability of timing between doses as well as the likelihood of breakthrough infections and resulting hybrid immunity makes for a much more heterogenous population (for which both the initial antibody levels and rate of waning of antibody levels are much less well characterised).

2) For data extracted from meta-analysis (especially TND), how did the authors deal with population natural infection histories in the study populations?

The primary studies from which data were extracted were focused on identifying vaccine effectiveness in vaccinated individuals compared to control (naïve) individuals. Thus, different attempts were made across these primary studies to identify previous infection (e.g. self-reported previous infection, documented previous infection in clinical database, seronegative to N antigen). These criteria used in the original studies may have been more or less effective in excluding previously infected individuals depending on the study. However, as stated in the methods, we report the effectiveness as listed in the primary study.

As discussed above, the likelihood of previous (undetected) infection became much higher after the omicron wave (as infection became much more widespread and population testing rates decreased significantly), and was one reason to maintain our original data cutoff date of 2 March 2022.

3) Line 86, need to explicitly describe the searching, screening and inclusion process. 15 studies seemed not representative for current literatures.

We thank the reviewer for this suggestion. As described above, we have now explicitly outlined our procedure in the supplementary methods. We agree that a cursory glance might suggest that there are many more reports of efficacy studies in the literature. However, many of these do not provide the data in a form suitable for analysis. For example, as now clearly indicated in Figure S9, 32 papers were excluded because they did not specify a time course of effectiveness after vaccination (therefore including a spectrum of waning responses), and 14 papers were excluded because the effectiveness estimates reported were over a number of successive waves due to different SARS-CoV-2 strains. We hope that our search, screening and inclusion criteria are now very clear and explicit as suggested.

4) Line 201 & 208 Correlation value should also be reported.

We have now included correlation values at line 201. Correlation values are not relevant at line 208.

5) Line 226-228 It's unclear how this analyses was performed. Isn't the correlates model originally fitted based on nAbs?

We now more clearly clarify that this section related to plotting of data extracted from identified papers on top of our original (unchanged) model from Khoury et. al. *Nature Medicine*, and this section was simply an overview of how well the extracted data agreed with the original model. We have now re-worded this section to make this clearer. The new text now reads:

Figure 3A shows the previously reported relationship between neutralising antibody titre and protection from symptomatic SARS-CoV-2 infection (red line and shaded 95% confidence intervals, reproduced from reference⁴). This line is overlaid with the 157 values of estimated neutralisation titre / reported effectiveness combinations we were able to obtain for protection against symptomatic infection from reported studies. Similarly, the 247 reported values for protection from severe COVID-19 are also plotted on top of our previous estimate

of the relationship between neutralising antibodies and protection from severe disease (Figure 3B).

6) Line 239 Correlation is for symptomatic and severe diseases combined? This study seemed to focus on V_e against severe COVID-19, so not sure why symptomatic was also included.

The reviewer is correct that we have presented correlations for both symptomatic and severe disease. While this manuscript is predominantly focusing on the relationship between neutralising antibodies and protection from severe disease, we also used this opportunity to confirm our previously identified relationship between neutralising antibodies and symptomatic disease, and show that it remains true over time and against multiple variants.

7) Lines 245-248. There are clear opposite real-world evidence suggested very high VEs against severe COVID-19 after one or two doses inactivated vaccines.

The reviewer suggests that the protection observed from inactivated vaccines may contradict the work presented here. We have now shown (above in response to point 2) that data from Ranzani et. al. 2022, agrees with the work presented here. In relation to the McMenamin ME 2022 *Lancet Inf Dis* study, we note that several issues made it difficult to include in this study:

- (1) This study does not look at decay of effectiveness (and reports on effectiveness over a > 14-month window),
- (2) this study does not report on a specified variant, (and covers the alpha, delta, and micron periods),
- (3) the study reports > 900,000 confirmed SARS-CoV-2 infections of which only 5,566 mild or moderate cases fulfil the criteria for inclusion while 8,875 (i.e. 60% more) severe or fatal cases are included.

However, although the time and variant-dependent neutralisation titres cannot be calculated for this study (because of factors (1) and (2) above), and therefore we cannot overlay data from this study onto panels A or B of Figure 3 from the main paper, we can still use the relationship between protection from symptomatic and severe infection to test whether this study shows higher protection against severe infection than would be expected. Plotting the McMenamin reported efficacy against Mild and Severe disease for one and two dose CoronaVac on our figure 3C (from the main paper) we see very good consistency with the model (black dots overlaid by black shading below are from McMenamin 2022 *Lancet Inf Dis*). Note that the black dot outside the confidence intervals of the model is actually the two-dose data point, which was reported by McMenamin has having lower efficacy against Mild protection than a single dose (though horizontal CIs are overlapping).

Therefore, we do not see obvious systematic evidence in the literature that data on inactivated vaccines would lead to grossly different results. Rather, so far we see these results support our existing findings. Since data on the efficacy over time for inactivated vaccines were not uncovered by our literature search, we have not included this in the manuscript directly however we have now included the following comment in the limitations section of the discussion:

We also note that the studies identified all reported on either mRNA or viral-vector vaccines, and we did not identify any reports of efficacy over time after vaccinating with inactivated vaccines. It would indeed be interesting to confirm whether this relationship is maintained for inactivated vaccines.

Reviewer #2 (Remarks to the Author):

This excellent paper extends earlier, highly influential publications from the research group on the role of humoral immunity in protection against SARS-CoV-2 infection and disease. The basis of the present paper is the use of published methods to further analyze the protective role of virus neutralizing antibodies (NAbs). A previous report established the normalized NAb titer as the dominant Correlate of Protection (CoP) against SARS-CoV-2 infection in vaccine trials. That finding was not only important, it made great sense based on long standing knowledge of how NAbs act. Now, the authors show that NAbs also protect against severe Covid-19, and at lower titers than are required to protect against infection. That too is an important finding, and it is also one that is consistent with fundamental knowledge of the immune response to viral pathogens. Additional analyses address the also key topic of virus variants with various degrees of NAb resistance; and the effect of the waning

of NAb titers over time. Overall, this paper is impressive in respect of its scope, its significance and its fundamental soundness.

I have no suggestions about how to modify the text.

We thank the reviewer for this positive review and strong support for our manuscript

The authors are, I assume, well aware that researchers studying T-cell immunity have a tendency to talk up what they work on. They often make the point that T-cells protect against viral disease, including Covid-19. However, my own read of the Covid-19 vaccine literature is that the hard evidence supporting these claims is thin - almost to the point of non-existence. The authors' work in this area is of particular importance in that context. If I may make so bold as to suggest the next area of analysis for the present research team, it is to look at the ever-increasing amount of T-cell response data available from the same vaccine trials they analyzed in the present paper. Is there a way to apply the same techniques? My prediction is that they would find scant evidence for T-cell responses being a CoP for Covid-19 vaccines, either in infection (no surprise) or disease (an eye-opener for some researchers).

We thank the reviewer for this suggestion and will consider this for future research. In addition, in light of the opinion of this reviewer, which is different to the comments made by reviewer 1, we have also included a discussion section on the relative importance of T-cells in protection from severe disease. This has been outlined above in response to reviewer 1's comments.

Reviewer #3 (Remarks to the Author):

The authors used data from previously published studies on SARS-CoV-2 vaccine effectiveness to firstly predict levels of neutralizing antibodies over time. They then demonstrate an association between these predicted levels of neutralizing antibodies and vaccine effectiveness against both symptomatic and severe outcomes. This an interesting study, and the authors have a good rational for why they are using modeling and combining data from different studies to determine if there is an association between neutralizing antibodies and vaccine effectiveness against severe covid-19. They use a number of different modeling techniques to test the robustness of their results with consistent findings. The conclusions are supported by the data presented and the main limitations are discussed.

We thank the reviewer for their positive comments.

However, in parts of the results sections it was challenging to follow exactly which modeling strategy was being used without referring the supplementary material. The authors report in the results section that they identified and extracted data on 15 studies, but there are only very sparse details either here or in methods section on how these studies where actually found, what literature did they search? What time period did they use? Which search terms? What studies were excluded & why? Did they perform a systematic review or how were the studies selected?

We thank the reviewer for their comment, and for identifying an aspect of our manuscript that was unclear. As mentioned above in response to a similar comment from reviewer 1, we have now performed a comprehensive review and made our systematic search and exclusion criteria explicit and have brought it in line with the PRISMA guidelines. This has been outlined in the supplementary materials and has been described in detail in our response to the editorial comments and to reviewer 1 above.

There is a footnote in the Supplementary that “†Denotes studies that included efficacy against confirmed infection without reporting on symptoms. These data were not included in the final analysis.” It appears next to 6 of the studies – so were only 9 studies included in the analysis? It is not clear which parts of the analysis these studies were excluded from.

We apologise that this was unclear in our earlier submission. In actual fact, all efficacy was against symptomatic infection or severe outcomes, thus any efficacy data that reported on infection only (without symptoms) was excluded from all the analysis. There were two studies (Bruxvoort et. al. *BMJ* and Tartof et. al. *Lancet*) that *only* included appropriate data on confirmed infection (i.e. effectiveness data against a specified variant and over time). These were originally listed in supplementary table 1 as being picked up by our screening, but the footnote was to indicate that the data was not included in the analysis as it did not meet the criteria. We have now provided additional details of our screening and inclusion criteria to make it clear that papers must have reported on either or both of symptomatic infection and severe disease against a specified variant over time since vaccination in order to be included in our analysis. This is made explicit in the supplementary methods, as outlined above in response to editorial comments reviewer 1. Thus, to be very clear, the total number of studies that finally contributed data points to our analysis is now 15 studies.

Figure 2 and Figures 3 for example reading the legend it appears to show data from 12 of the studies (with Skowronski split in two)?

We thank the reviewer for pointing out this oversight with the Skowronski et. al. data. It should not have been split in two and should have been represented with only one symbol. This has now been changed.

A flow chart and perhaps referring to the PRISMA guidelines may help with this

We thank the reviewer for this suggestion. As outlined above in response to editorial comments and reviewer 1, we have now repeated our literature search and brought it in line with PRISMA guidelines for a systematic review and included the PRISMA specified flow chart detailing the search results, inclusions and exclusions.

For the first part of the paper, neither the introduction nor methods clearly explain why vaccine regimen, time since vaccination and variant are the key parameters of interest or whether there were any other variables considered that did not make it into the final model/ study selection?

We apologise that the reasoning for this was unclear. The initial studies of vaccine immunogenicity and protection were carried out against the ancestral variant, in the first 2-3

months of infection. Subsequently, it has become clear that waning neutralisation titres over time and the drop in neutralisation titre against SARS-CoV-2 variants have a major influence on neutralising antibody titres. Thus, to stress test the original model we sought to ascertain whether neutralising antibody titres remain protective under these circumstances. Thus, since vaccine regimen, time since vaccination and variant are key parameters that have been identified to affect neutralising antibody levels, these were considered to be key parameters for our analysis. We have now added a sentence towards the beginning of the results to explain why these are key parameters of interest. The relevant sentence reads:

These are key parameters of interest, as they have each been shown to independently influence neutralising antibody levels^{4,14,16}, and hence may each have independent impacts on protection.

Minor comments:

Title: The title should be revised to state what type of study it is

We thank the reviewer for this suggestion and have changed it accordingly. The title is now:

Predicting vaccine effectiveness against severe COVID-19 over time and against variants: a meta-analysis

Introduction line 51: The authors state: ‘Studies of the relationship between neutralising antibodies and protection from symptomatic SARS-CoV-2 infection...’ but they only reference one of their own papers in this sentence. Suggest to be rephrased to ‘we have previously...’ or add additional references for clarity

Thank you, additional references have now been added.

Introduction line 80: Suggest the sentence is revised to outline the study aims and hypothesis rather than stating the results (these statements should be reserved from the results and discussion).

We thank the reviewer for this suggestion, and have now changed the final two sentences of the introduction. They now read:

Using a previously published relationship between neutralising antibodies and protection we test whether knowing the vaccine regimen, time since vaccination, and SARS-CoV-2 variant, allows us to predict vaccine effectiveness against severe COVID-19. This will provide evidence as to whether changes in neutralising antibodies over time and against different variants are predictive of changes in vaccine effectiveness against severe outcomes.

Consider moving table S1 to the main manuscript, could move Figure1 to the supplementary instead

We have now moved what was previously table S1 to the main manuscript. We note that it is a very large table, and we are willing to follow editorial advice on whether this table is best left in the main text or supplementary materials.

Results line 116, the authors state that vaccine effectiveness was lower with the Delta variant compared to pre-Delta however, the confidence intervals overlap 1.

Correct.

Figure 2. There is a typo in the footnote

Corrected.

Reviewer #4 (Remarks to the Author):

This is a very important and innovative paper that uses available data to examine the relationship between covid-19 vaccine type and timing protection against severe disease from different variants. Then, the authors use a model of neutralizing antibody titers to predict the relationship with variant-specific protection against severe outcomes. It is a logically sound approach, but, as the authors acknowledge, it combines two relationships - first that efficacy wanes over time in a variant- and vaccine-specific manner, that neutralizing antibodies are a predictor of vaccine effectiveness and that neutralizing antibodies wane over time. Then, they essentially combine these relationships and assert that neutralizing antibodies can predict efficacy. Generally, it is a well-performed analysis that fills a knowledge gap, but does require a strong assumption of a causal relationship of vaccine -> neuts -> efficacy/effectiveness.

We thank the reviewer for their positive comments on our work.

I do have some specific concerns about how the studies were selected and analyzed.

1. This is a systematic review but the literature search is not presented in sufficient detail to understand the inclusion and exclusion criteria and how the search strategy was implemented. I strongly suggest that the PRISMA guidelines be followed.

Thank you for this comment, in line with this comment, and comments from other reviewers, we have now repeated the literature search and brought it in line with PRISMA guidelines for systematic reviews and revised the manuscript to include details of this literature search. We have outlined this in detail in the supplementary materials, along with a flow chart as recommended by the PRISMA guidelines. We thank all reviewers for their suggestions regarding this, as we feel it has greatly improved the quality and impact of this work.

2. The traditional checks that are performed in a meta-analysis were not done or not reported. A major question is whether there is too much heterogeneity between the studies for them to be combined. The substantial number of outliers suggest that may be the case. I suggest a formal analysis of I^2 or other such metrics.

We thank the reviewer for this suggestion, and for identifying that we should indeed have accounted for potential variations in reported effectiveness estimates between different studies. We have now accounted for this potential source of heterogeneity by implementing a mixed effects regression model. We have included the study from which the data was extracted as a random variable in this mixed effects model.

This work was different from a traditional meta-analysis, in that we were not attempting to summarise results, or find an overall effect, rather, we were aiming to determine whether results from the literature were compatible with our previously published model. Indeed, given the results of our previous modelling work (Khoury et. al. *Nature Med*) we have a strong prior that the relationship between neutralising antibody titres and vaccine effectiveness is both non-linear and related to a combination of vaccine, variant and time since vaccination.

Therefore, we elected to use a mixed effects model from the outset, which would be the recommended step should there be a high level of heterogeneity between studies. The random effect parameter ζ_{study} is then the most suitable measure of between study heterogeneity, as it accounts for heterogeneity after also accounting for the temporal, vaccine specific and variant specific, differences in efficacy.

In addition, the random effect ζ_{study} can be compared to the other parameters of the model, to aid in understanding the relative contribution of the unexplained heterogeneity between studies. From Table S1, we see the standard deviation of the random effect ζ_{study} is of the same order of magnitude as the vaccine and variant specific effects, and is also similar to the change in efficacy over a 3 month period. This emphasises the need for greater consistency in reporting of the timeframe, vaccines and the variants to which efficacy is reported, in order to reduce both within and between study heterogeneity in effectiveness estimates (and is another reason why we only included efficacy estimates for identified vaccines / variants over a specified time frame following vaccination).

The mixed effects model is now detailed in the methods section of the manuscript. The relevant section in the reads:

Mixed Effects Model Fitting

To determine if vaccine effectiveness against severe COVID-19 was dependant on vaccine, variant and/or time since vaccination we fit mixed effects model to vaccine effectiveness with vaccine and variant as categorical covariates, time as a continuous covariate and included a random effect for the study from which the data came. The model was:

$$Eff = Eff_{base} - A_i - B_j - C_j t + \zeta_{study} \quad \text{Equation 1}$$

Where A_i is a vaccine specific adjustment for vaccine i , B_j is a variant specific adjustment for variant j , C_j is a variant specific parameter determining the change in effectiveness over time since vaccination (t) and ζ_{study} is a random effect for the study from which the data came. Values of these parameters are given in Table S1.

Minor comment:

Fig 3. The different symbols for each study type are not helpful since they are so small and largely clustered in one region of the graph.

We thank the reviewer for pointing out the difficulties identifying individual data points in the main figure. We have now added supplementary figures that show a breakdown of the data by study, where the data are presented much more clearly (Supplementary Figures 5-8).

REVIEWER COMMENTS

Reviewer #1 (Remarks to the Author)

I appreciate of the authors revisions to some my comments (Reviewer 1), while I'm afraid that my most concerning comments were not fully addressed (with two comments not replied at all).

The major concern of this study is that it is likely to provide biased (and could be misleading) predictions when generalizing (in particularly extrapolating as the model was mostly informed by high level antibodies) the model to predict vaccines effectiveness neutralizing antibodies levels were low. Several issues (see fowling listed points for details) in this study could contribute to this bias, while the authors seemed not bothered to address any.

- 1) The model is fitted to mainly mRNA vaccines, which induced high levels of antibodies against ancestral strains. It means that the model was not informed by lower levels antibodies, and therefore the model fitted "association between neutralizing antibody and vaccine effectiveness" at low antibody levels were actually extrapolation of such association at high antibody levels. As a result, even the authors stated the high correlation between predicted and observed vaccine effectiveness against severe diseases (which however may not be as good as they stated but will discuss later), such findings could only be confident to be true when neutralizing antibodies are high (which is no longer applicable given the substantial immune escape of Omicron).

I generated the following figure using the data and codes provided by the authors as well as the literatures about effectiveness against Omicron infections after 2-dose inactivated (mostly Coronavac) vaccines that were excluded by the authors. This figure again supports my concern about the misestimation of vaccine effectiveness when antibody levels are low. None of the point estimates of VE against severe diseases from real-world observations fell in the confidence intervals that predicted by the authors. In addition, for those the predicted confidence intervals barely covered the lower boundary of VE from observations, these estimates were derived from the elderly who tended to have even lower antibody levels (thus even lower predicted VE by the authors).

- 2) The statistics that the authors used to support the high accuracy of their predictions were suspiciously selective. As I raised in my previous comments, this study focused on predicting VE against severe infections, but the high correlations of predicted and observed symptomatic infections were used here and there to support their claims (e.g., in abstract they reported a correlation of 0.9 but is actually for symptomatic protection).

The authors stated that “..the real-world data points maintain the predicted relationship between symptomatic and severe protection (Spearman’s correlation = 0.7)..”. It looks highly correlated and suggested a good model performance at the first glance; however, this correlation is actually between prediction of symptomatic and severe protection. Strictly speaking, the authors swapped the concepts and measurements for predicting protection of VE against severe infections. It’s worth noting that two measurements can be perfectly correlated but at the same time symmetrically biased.

Similar issues happened when the authors used the association between predicted symptomatic infections and severe infections to back up their model predictions to my challenge of the McMenamin study. As I showed in my previous plot, the confidence intervals of predictions and observations of VE against severe infections barely overlap except for those elderly people.

In addition, the authors replied that “9 of the 11 new points in the severe correlation having overlapping confidence intervals with the model (panel B of figure below)”. However, when have a closer look, 5 are actually from mRNA vaccines (which is expected to have good agreements), while the rest 6 estimates from CoronaVac (the

most concerning one) were systematically higher than the authors predictions. In other words, some of the good performance reported by the authors were informed by those expected-to-be good while masked those real concerns and risks when applying the model.

- 3) The strict inclusion criteria of studies that used to validate the model performance could limit the model prediction power and the generalizability of their model. The authors included 12 studies and excluded 96 studies, of which almost one third ($n = 33$) was due to no time course since 2nd infection. These studies could be included to further assess the robustness of their model, as the model predictions implicitly provide the highest possible vaccine effectiveness (after 14 days). The vaccine-induced antibody decayed with time and so did the VE (according to the authors hypothesis), and therefore VE from observational studies is not expected to be higher than what predicted. While the authors chose to tighten the inclusion criteria, they coincidentally excluded studies that challenged their model performance (indicated in the first figure).

Regarding the systematic review, I appreciate the authors provide the PRISMA diagram in their revision. However, given that the authors provided no search term and no search period, there is impossible to assess the quality of their search strategy. In addition, the authors are also recommended to provide the PRISMA reporting list.

- 4) The underlying assumption by the authors is that the correlation of protection against severe outcome is associated with antibodies. Yet evidence suggested the importance of T-cell immunity in facilitating antibody generation and functioning and provide protective mechanisms. The authors complained me not providing literatures, while I see this as the failure of the authors to understanding their research fields. I here attached several excellent reviews and studies for the authors references [PMID: 34017137, 35013199, 33497610, 32979941, 31257567, 33261718, 32555388, 32991844, 33408181].

There are more and more evidencing suggesting the decoupling of antibody against severe infections. Despite from those VE studies from Hong Kong Omicron wave, laboratory studies found neutralizing antibodies against Omicron after 2 doses of BNT162b2 or 2 doses of CoronaVac with undetectable titers (PMID: 35675370), yet

these vaccine statuses still provide high protection against severity as predicted by antibody levels.

Last but not least, the authors replied by quoting the other reviewer as below:

We note that the comment from Reviewer 1 is in direct contrast to the comment from Reviewer 2 below, who noted that “researchers studying T-cell immunity have a tendency to talk up what they work on. They often make the point that T-cells protect against viral disease, including Covid-19. However, my own read of the Covid-19 vaccine literature is that the hard evidence supporting these claims is thin – almost to the point of non-existence.” These two very divergent views on the role of T-cells are indeed representative of the views of the scientific community at large (and the strong feelings of many on this topic), and therefore it is indeed important to address this question, which we have now done in the discussion.

In my opinion, this is a dangerous mindset in performing studies and drawing conclusion. Absent of evidence is not the evidence of absent. Ignoring contributions from other unmeasured immunizes may lead the field to overly focus on antibody while reducing exploration on other proactive mechanisms as stated here (PMID: 35324269). I am actually not studying T cell immunity, but an epidemiologist working with serological data and understand of the limitations of neutralizing antibodies in antigenically variable pathogens.

Reviewer #3 (Remarks to the Author):

The authors have suitably revised the manuscript. Their search criteria and methodology and now easier to follow and much better explained.

Reviewer #4 (Remarks to the Author):

All my concerns have been sufficiently addressed.

Response to Reviewers' Comments

Reviewer #1

I appreciate of the authors revisions to some my comments (Reviewer 1), while I'm afraid that my most concerning comments were not fully addressed (with two comments not replied at all).

We apologise if the reviewer feels we did not address some of their comments. We have looked through our response and unfortunately we cannot find the comments that the reviewer feels did not receive a reply.

The major concern of this study is that it is likely to provide biased (and could be misleading) predictions when generalizing (in particularly extrapolating as the model was mostly informed by high level antibodies) the model to predict vaccines effectiveness neutralizing antibodies levels were low. Several issues (see fowling listed points for details) in this study could contribute to this bias, while the authors seemed not bothered to address any.

We thank the reviewer for their comments. It is correct that the original model of correlates of protection in COVID-19 (Khoury et al, Nature Medicine 2021) was parameterised using data with relatively high neutralisation titres. Indeed, this model was parameterised with the Phase 2 and Phase 3 data of the first seven vaccines approved for use in COVID-19, none of which had an efficacy of <50% against symptomatic infection, or <85% against severe COVID-19 disease. Therefore, the lower half of the model was indeed an extrapolation. However, it is important to stress that the current work does not 'provide predictions'. Instead, it tests the earlier predictions around severe infection made in the Khoury et. al., Nature Medicine 2021 paper against new data that has since become available. Thus, the current study represents an important validation that the extrapolation in the original model was remarkably accurate at least for the mRNA and viral vector vaccines that had sufficient data to allow us to capture them in this systematic review. That is, manuscript Figure 3A and B show that the published effectiveness data at low neutralisation titres is very consistent with the predictions of the earlier model. We have now added an additional section to the discussion section that addresses this concern. The relevant section of the discussion now reads:

Although the association between neutralising antibodies and protection from symptomatic SARS-CoV-2 infection has been investigated in several settings^{4, 5, 6, 7, 14, 16}, protection from severe infection has heretofore remained more difficult to unravel. Our first analysis of the relationship between neutralising antibodies and protection from COVID-19 was parameterised based on the Phase 2 (immunogenicity) and phase 3 (efficacy) data from seven vaccines and from convalescent individuals⁴. However, none of these studies reported a protective efficacy below 50% for symptomatic infection, or below 85% for severe COVID-19. Thus, the model estimates of efficacy at low neutralising antibody titres were an extrapolation from the data available at the time. In addition, all of the phase 2 and phase 3 licensure studies reported responses and protection against the ancestral virus in the first months after vaccination. Here we aggregate the available epidemiological data to investigate whether the relationship between neutralising antibodies and protection from severe COVID-19 remains predictive across a diverse range of real-world scenarios of different vaccines, variants, and time since immunisation. Our analysis demonstrates that reported changes in neutralising antibodies over time and against different variants are indeed predictive of changes in protection from severe COVID-19 across these different scenarios at least for the vaccines captured in our systematic review (Figure 3). It is notable that the previously developed model appears highly predictive of protection at

low neutralising antibody levels, in the presence of waning immunity and immune escape variants (Figure 3A and B), providing an important validation of the model. This provides strong, though indirect evidence that neutralising antibodies are a correlate of protection from severe COVID-19.

1) The model is fitted to mainly mRNA vaccines, which induced high levels of antibodies against ancestral strains. It means that the model was not informed by lower levels antibodies, and therefore the model fitted “association between neutralizing antibody and vaccine effectiveness” at low antibody levels were actually extrapolation of such association at high antibody levels. As a result, even the authors stated the high correlation between predicted and observed vaccine effectiveness against severe diseases (which however may not be as good as they stated but will discuss later), such findings could only be confident to be true when neutralizing antibodies are high (which is no longer applicable given the substantial immune escape of Omicron).

We apologise that there seems a slight confusion between (a) the *correlation* between predicted neutralising antibodies and protection and (b) the *modelling* of this relationship.

To better explain, the *correlation* between predicted neutralising antibodies and protection does not depend on the fitted model associating neutralisation and protection. To determine this *correlation* we performed a Spearman rank correlation (i.e. a non-parametric correlation, and therefore one that is not related to any actual values or model). The results of this correlation test demonstrate that predicted neutralising antibody titres are highly correlated with protection from both symptomatic ($\rho = 0.95$, $p < 0.001$) and severe ($\rho = 0.72$, $p < 0.001$) COVID-19 disease. This correlation is entirely independent of the Khoury et. al. model discussed later in the manuscript.

With respect to the Khoury et. al. *model*, in the original study this was fitted to data from the phase 1 and phase 2 studies of seven different vaccines and also data on convalescent subjects (eight datapoints in total) (reference 4). Two of the vaccine studies were of mRNA vaccines, one was protein based, three were viral vector vaccines and one was an inactivated vaccine. So the original model was not “*fitted mainly to mRNA vaccines*”.

Importantly, the existing manuscript does not include any model fitting. Rather, the observed data and estimated neutralising antibody titres (obtained in the current study) are simply plotted on top of the previously published model to test whether it remains predictive of protection in the context of waning antibodies and new SARS-CoV-2 variants. As the reviewer outlines below, the full data and code for this model have been freely available online since July 2021 (and indeed have been used by multiple regulators to inform vaccine approval, by public health agencies to predict optimal vaccine policies, and possibly by the reviewer to generate their figure).

To clarify this potential misunderstanding in the text, we have added a section to the end of the results section titled “Correlation between neutralising antibody titre and vaccine effectiveness against severe COVID-19” that reads:

We note that this observed correlation between estimates of neutralising antibody titres and effectiveness is independent of the model developed by Khoury et. al.⁴. Rather, once published neutralising antibody titres are adjusted to account for (i) immune waning and (ii) drop in recognition of the circulating variant, the reported vaccine effectiveness is remarkably well correlated with these neutralising antibody titres.

In addition we have added an extra section to the start of the discussion that reads:

One challenge in determining whether neutralising antibody titres are associated with protection from severe COVID-19 is that the predicted 50% protective titre is below the limit of detection for many in vitro neutralisation assays^{4, 32, 33}. However, by adjusting reported neutralising antibodies to incorporate the effects of immune waning and recognition of circulating variants and then correlating these with observed protection, we show that predicted neutralising antibody titres are strongly correlated with reported estimates of protection against severe COVID-19 disease (Spearman's $\rho=0.72$, $p<0.001$).

We have also added a later section to the discussion that reads:

In addition, we previously identified a similar relationship between neutralising antibodies and protection from severe COVID-19, although this was based upon a small amount of data on severe infection⁴. Importantly, the lowest reported titres were around 20% of convalescent antibodies, and so predicted efficacies against severe disease for neutralising antibodies below this level are based on extrapolation only.

I generated the following figure using the data and codes provided by the authors as well as the literatures about effectiveness against Omicron infections after 2-dose inactivated (mostly Coronavac) vaccines that were excluded by the authors. This figure again supports my concern about the misestimation of vaccine effectiveness when antibody levels are low. None of the point estimates of VE against severe diseases from real-world observations fell in the confidence intervals that predicted by the authors. In addition, for those the predicted confidence intervals barely covered the lower boundary of VE from observations, these estimates were derived from the elderly who tended to have even lower antibody levels (thus even lower predicted VE by the authors).

We thank the reviewer for clarifying their concerns about the work and apologise for any confusion caused.

The reviewer raises two concerns here.

Concern 1: That we may have “excluded” data that negates the results of the paper:

The criteria for inclusion in the study were uploaded on the original version of the preprint on medRxiv on 9 June 2022 and appeared freely available online from 14 June 2022 (<https://www.medrxiv.org/content/10.1101/2022.06.09.22275942v1>). As helpfully suggested by multiple reviewers, we have since formalised our systematic search according to PRISMA guidelines, while retaining these same inclusion criteria. This captured an additional 2 papers. We note that the systematic search still did not include any of the studies suggested by the reviewer. The reason that no inactivated vaccines were included in the analysis was because they did not meet the inclusion criteria for our systematic review. For example, of the studies suggested by this reviewer, all except one did not contain a time course of effectiveness / efficacy measurements. This requirement of having a time course of effectiveness is absolutely essential to the design of our study, since a major aspect of our work relates the decay of neutralising antibodies to the decay of efficacy to understand if they continue

to be correlated. The final study suggested by this reviewer was published after the date cutoff for our systematic review. The reviewer may be concerned that we have changed the search terms or cutoff dates to avoid particular papers, but the search terms and the cutoff date have remained constant and are clearly stated in the medRxiv upload of June 9th 2022

(<https://www.medrxiv.org/content/10.1101/2022.06.09.22275942v1>).

Concern 2: Data from inactivated vaccines invalidate the conclusions of the paper

Again, it is important to differentiate between the *correlation* between neutralising antibody titre and protection, and the *model* of this.

With respect to the correlation, this is independent of any model and relies on a simple non-parametric Spearman correlation. In order to test the reviewer’s concern, we extracted the data from the figure included by the reviewer to see the effect on the Spearman correlation between predicted neutralising antibodies and protection from severe COVID-19. (Note that we would disagree with exactly where the points are positioned, as discussed further below, but for this analysis we used the exact points indicated by the reviewer).

With respect to the correlation between predicted neutralising antibodies and protection from severe SARS-CoV-2 infection, we found:

	Spearman’s Rho	p-value
Using original data only	$\rho=0.72$	<0.001
With reviewer data included	$\rho=0.75$	<0.001

Thus, with respect to the question “*would inclusion of the reviewer-selected inactivated vaccine effectiveness data invalidate the conclusion that neutralising antibodies are predictive of protection from severe disease?*”, the answer is that it would not. In fact, inclusion of the additional data suggested by the reviewer strengthens the conclusion that neutralising antibody titre is correlated with protection from severe COVID-19.

With respect to the visual overlay of the data onto the published model by Khoury et. al., we would make two points. Firstly, we are not convinced that the studies plotted by the reviewer are truly representative of the effectiveness of inactivated vaccines. To check this, we did our own, very brief (not systematic) search for estimates of effectiveness of inactivated vaccines against severe COVID-19, and present the results of this brief search below in Figure 1. We found estimates of effectiveness against severe COVID-10 ranging from 0% up to 82% (in contrast to the studies plotted by the reviewer, which all have severe effectiveness estimates of greater than 56%). The data we extracted was from Paternina-Caicedo et. al. 2022, <https://doi.org/10.1016/j.lana.2022.100296>, Suah et. al. 2022, <https://doi.org/10.1080/22221751.2022.2072773>, and Premikha et. al. 2022, <https://doi.org/10.1093/cid/ciac288>. We note also that in the previous round of review the reviewer mentioned the paper by Ranzani et. al. (BMJ, 2021, <https://doi.org/10.1136/bmj.n2015>) but did not include this in their figure – we have also included that data in figure 2 below (plotted in purple).

In addition, the extra studies we uncovered were estimates for inactivated vaccine effectiveness against Delta and pre-Delta variants, for which one would expect *higher* effectiveness than in the studies plotted by the reviewer (which were predominantly against the Omicron variant). Thus, it appears that

the data placed on the figure by the reviewer may not be representative of the published data on effectiveness of inactivated vaccines.

Figure 1 Estimates of inactivated vaccine effectiveness against severe SARS-CoV-2 infection extracted from studies selected by the reviewer (red) and in our brief search (blue) and by the reviewer in round one of reviews, but not plotted on the figure (purple). Extracted data are groups by studies that either were included in the reviewer's figure (left) or were not included in the reviewer's figure (right).

Secondly, as mentioned above, we are not convinced that the reviewer has placed the points shown in their figure in the correct position on the x-axis, as we believe their assumption about the neutralising antibodies conferred by Coronavac vaccination is based only on very early data (which was also what was reported in the Khoury et. al. Nature Medicine 2021 paper). The original phase 2 and phase 3 studies of Coronavac used a two-week dose spacing. After the low efficacy of this regime was observed, the recommended dosing was extended to four weeks. It has been shown that that this small delay resulted in a doubling of neutralisation titres, regardless of the dose given (Figure 2 below, data taken from Xin et. al. Nature Comms 2022 <https://doi.org/10.1038/s41467-022-30864-w>). In addition, the real-world roll out of the vaccine at times led to longer spacing of doses (which are not specified in many publications). Thus, it can be quite difficult to know the neutralisation titre after Coronavac vaccination (and therefore where to place these points on the x-axis) and a more detailed analysis would be required for this.

Figure 2 Neutralising antibody titres measured after a 2-week (14 day) dosing interval (red) or a 4-week (28 day) dosing interval (blue) using data from Xin et. al. Nature Comms 2022 (<https://doi.org/10.1038/s41467-022-30864-w>)

We do believe that the concerns raised by the reviewer necessitate some commentary, and for this reason we have added a section in the discussion that recognises that based on this analysis it is not possible to draw a conclusion on whether the association between neutralising antibody titres and protection from severe disease holds for inactivated vaccines, as no studies of inactivated vaccines met the inclusion criteria for our systematic review, and that this is an area that warrants future research. The relevant section of the discussion now reads:

We also note that the studies identified in our systematic review all reported on either mRNA or viral-vector vaccines, and we did not identify any reports of efficacy over time after vaccinating with inactivated or protein-based vaccines. Some studies of vaccine effectiveness of inactivated vaccines against the omicron variant suggest that effectiveness may be higher than expected due to neutralising antibodies alone^{45, 46}, although these studies did not meet the criteria for our systematic review as they were published after the date cut-off. Further analysis is required to determine whether the correlation between neutralising antibodies and protection against severe disease is different in some way for inactivated vaccines.

2) The statistics that the authors used to support the high accuracy of their predictions were suspiciously selective. As I raised in my previous comments, this study focused on predicting VE against severe infections, but the high correlations of predicted and observed symptomatic infections were used here and there to support their claims (e.g., in abstract they reported a correlation of 0.9 but is actually for symptomatic protection).

We are sorry the reviewer feels the representation of our results is imprecise, as we felt we had been transparent and open throughout the manuscript as to the methods and statistics used. Indeed, in the abstract we actually reported correlations for both symptomatic and severe disease, however it was perhaps unclear to the reviewer, as correlations appeared next to each other. The abstract previously read:

We find that predicted neutralising antibody titres are strongly correlated with observed vaccine effectiveness against symptomatic and severe COVID-19 (Spearman $\rho = 0.95$ and 0.72 respectively, $p < 0.001$ for both)

However, given the reviewer's confusion we have now modified it to read:

We find that predicted neutralising antibody titres are strongly correlated with observed vaccine effectiveness against symptomatic (Spearman $\rho = 0.95$, $p < 0.001$) and severe (Spearman $\rho = 0.72$, $p < 0.001$) COVID-19

The authors stated that “..the real-world data points maintain the predicted relationship between symptomatic and severe protection (Spearman's correlation = 0.7)..”. It looks highly correlated and suggested a good model performance at the first glance; however, this correlation is actually between prediction of symptomatic and severe protection. Strictly speaking, the authors swapped the concepts and measurements for predicting protection of VE against severe infections. It's worth noting that two measurements can be perfectly correlated but at the same time symmetrically biased.

Similar issues happened when the authors used the association between predicted symptomatic infections and severe infections to back up their model predictions to my challenge of the McMenamin

study. As I showed in my previous plot, the confidence intervals of predictions and observations of VE against severe infections barely overlap except for those elderly people.

We apologise for any confusion. The reviewer is concerned that we “*swapped the concepts*” and that the stated relationship is “*is actually between prediction of symptomatic and severe protection*”. We had felt that this methodology and results were explained in the text, yet we apologise if our wording was unclear to the reviewer. In the previous version we had realised the potential for confusion and tried to make this concept clear. For example, we placed this in an entirely new section titled “*Effectiveness against symptomatic infection predicts effectiveness against severe COVID-19.*” Within which we stated the below (our emphasis)

*Therefore, we next sought to assess the utility of the published model of correlates of protection from severe COVID-19 using an approach that did not rely on estimating neutralising antibody titre. The published correlates model⁴ explicitly predicts a (non-linear) relationship between protection from symptomatic disease and protection from severe disease (red line in Figure 3C). **That is, for any observed level of protection from symptomatic infection, the published model implicitly predicts a corresponding level of protection from severe COVID-19.** This has the major advantage of being independent of any assumptions of the underlying neutralising antibody titres.*

However, to clarify further we have now also added the following towards the end of the section titled “*Effectiveness against symptomatic infection predicts effectiveness against severe COVID-19.*”

It is important to note that this analysis, unlike the analysis presented in the previous section, does not directly estimate the correlation between neutralising antibodies and protection. Rather, it tests the model’s prediction of the relationship between protection from symptomatic infection and protection from severe infection (based on the relationship of each to neutralising antibody titre).

In addition, the authors replied that “9 of the 11 new points in the severe correlation having overlapping confidence intervals with the model (panel B of figure below)”. However, when we have a closer look, 5 are actually from mRNA vaccines (which is expected to have good agreements), while the rest 6 estimates from CoronaVac (the most concerning one) were systematically higher than the authors predictions. In other words, some of the good performance reported by the authors were informed by those expected-to-be good while masked those real concerns and risks when applying the model.

The reviewer argues that the data analysed in the previous response to reviews included mRNA vaccine data. Here it is important to point out that this analysis was carried out in response to the previous comments by this reviewer, who cited 4 papers as examples of work that they believe had omitted from our analysis. We analysed the suggested papers and found that while none of these actually met the criteria for our systematic review, 2 of them were only outside of the date cut-off and met all other criteria. We therefore analysed these 2 papers and extracted the data available (the 11 points to which the reviewer refers). Since these were papers specifically chosen by the reviewer as examples that they felt had been omitted and may therefore negate our analysis, we found it encouraging that when analysed in detail, they actually support the analysis. The reviewer is concerned that our analysis of their suggested papers in some way “*masked those real concerns and risks*”. However, we were only analysing the data the reviewer put forward as examples of studies that contradicted our results. We did not choose those papers.

In this round of reviews, the reviewer has listed some additional papers as examples that they feel contradict our conclusions. However, as we have detailed above, none of these studies suggested by the reviewer negate the conclusion that neutralising antibodies are *strongly correlated with protection against severe disease*. This is despite the fact that the studies did not fall within the criteria for the systematic review. Moreover, as we point out above, the different dosing regimen used in the clinical trials versus in the field make it difficult to estimate neutralising antibody titres for Coronavac, and more analysis is required to determine the neutralising antibody titres associated with these points (and therefore to truly assess whether or not they align well with the *model*).

However, as requested by the reviewer we now include a specific comment referencing the data on inactivated vaccines, some of which has been used to argue for non-neutralising effects:

The relevant paragraph of the discussion now reads:

We also note that the studies identified in our systematic review all reported on either mRNA or viral-vector vaccines, and we did not identify any reports of efficacy over time after vaccinating with inactivated or protein-based vaccines. Some studies of vaccine effectiveness of inactivated vaccines against the omicron variant suggest that effectiveness may be higher than expected due to neutralising antibodies alone^{45, 46}, although these studies did not meet the criteria for our systematic review as they were published after the date cut-off. Further analysis is required to determine whether the correlation between neutralising antibodies and protection against severe disease is different in some way for inactivated vaccines.

3) The strict inclusion criteria of studies that used to validate the model performance could limit the model prediction power and the generalizability of their model. The authors included 12 studies and excluded 96 studies, of which almost one third (n = 33) was due to no time course since 2nd infection. These studies could be included to further assess the robustness of their model, as the model predictions implicitly provide the highest possible vaccine effectiveness (after 14 days). The vaccine-induced antibody decayed with time and so did the VE (according to the authors hypothesis), and therefore VE from observational studies is not expected to be higher than what predicted. While the authors chose to tighten the inclusion criteria, they coincidentally excluded studies that challenged their model performance (indicated in the first figure).

We understand that the reviewer is concerned that the restriction in our systematic review that papers must “*present a time series of efficacy/effectiveness*” is overly stringent. We found this criteria to be particularly important for the analysis presented in this manuscript as one of the key premises of the study is to look at whether the changes in neutralisation titres (over time) and to different variants can explain the changes in observed effectiveness. In addition, we found that many studies aggregate efficacy results over extended periods (six to twelve months is common), where we expect significant differences in neutralising antibody titres and protection due to the effects of waning immunity. Moreover, the proportion of people vaccinated at different times is usually not specified. Therefore, from conception, we chose to focus not on obtaining every single study that reported an effectiveness, often aggregated over a considerable block of time, but rather on studies with the highest quality data on vaccine effectiveness, namely those which stratified effectiveness estimates over time.

The reviewer is concerned that we may have tightened our criteria in our revision, and thereby omitted particular articles. As indicated above, the criteria did not change when we extended our search to

become more systematic (as requested by the reviewer). Moreover, our search criteria were motivated by data quality rather than by a preconceived desire for a particular result.

Regarding the systematic review, I appreciate the authors provide the PRISMA diagram in their revision. However, given that the authors provided no search term and no search period, there is impossible to assess the quality of their search strategy. In addition, the authors are also recommended to provide the PRISMA reporting list.

We apologise for this oversight. Due to a technical issue, the search criteria, which we originally included in our response to reviewers, were unintentionally removed before final submission to reviewers but were included in the submission to the editor only. The paragraph on the search criteria which was unintentionally sent to the editor only reads:

In order to identify studies to be used in this meta-analysis, we searched PubMed for papers indexed between inception and 2nd March 2022 (PubMed search: (SARS-CoV-2 OR COVID-19) AND (followup OR waning OR duration OR durable) AND (protection OR efficacy OR effectiveness)) and also monitored other public sources of information such as Twitter and medRxiv.

We have now also included this paragraph as the opening paragraph of our supplementary methods.

4) The underlying assumption by the authors is that the correlation of protection against severe outcome is associated with antibodies. Yet evidence suggested the importance of T-cell immunity in facilitating antibody generation and functioning and provide protective mechanisms. The authors complained me not providing literatures, while I see this as the failure of the authors to understanding their research fields. I here attached several excellent reviews and studies for the authors references [PMID: 34017137, 35013199, 33497610, 32979941, 31257567, 33261718, 32555388, 32991844, 33408181].

We thank the reviewer for sharing their opinion. We note that the purpose of this work is to directly test whether neutralising antibody levels are significantly correlated with protection from severe SARS-CoV-2 disease (which they are, $\rho=0.72$, $p<.001$). This work does not set out to or claim to look at other immune mechanisms. There is no doubt that T follicular helper cells (for example) are key to the generation of antibody responses. However, our work asks whether neutralising antibody levels are predictive of subsequent clinical outcome. It casts no view on whether B cells, T cells, NK cells, or other mechanisms may or may not be important.

As outlined in our discussion, there is strong independent evidence that antibodies are mechanistic in protecting from severe SARS-CoV-2 infection. Multiple studies show that administration of monoclonal antibodies during symptomatic SARS-CoV-2 infection reduces the risk of subsequent hospitalisation – demonstrating a direct antibody effect in reducing the severity of infection (independent of their role in protection from acquisition of infection). To address the reviewer's concerns we have modified the text as below.

In this study we confirm that the relationship between neutralising antibodies and protection from severe COVID-19 is maintained as neutralising antibody titres change over time and against specific SARS-CoV-2 variants. Passive antibody studies have also directly demonstrated a mechanistic role for antibodies in protection from severe COVID-19. Administration of antibodies during symptomatic SARS-

CoV-2 infection can reduce the risk of subsequent hospitalisation or death by up to 85%⁴⁰ (reviewed in⁴²). This demonstrates a direct role for antibodies in reducing infection severity, independent of their role in preventing acquisition of infection. However, this does not prove that antibodies are exclusively responsible for protection against severe disease, and we cannot exclude the possibility that there are alternate mechanisms, such as T-cells, that also contribute to protection. Some evidence has suggested a potential role for T cell responses when neutralising antibody responses are not detected⁴¹ However, since cellular responses and neutralising antibodies typically correlate it is difficult to determine whether T cells are causal in this instance or merely correlated with a level of neutralising antibodies that is below assay detection⁴². In addition, since T cell help is required for the generation of high titre neutralising antibody responses, they likely play an important indirect role in protection. Therefore, while we can conclude that neutralising antibodies are associated with protection from severe disease (this study), and that passively administered antibodies can reduce severe diseases⁴², more work is still required to determine the contribution of cellular immune responses to protection.

There are more and more evidencing suggesting the decoupling of antibody against severe infections. Despite from those VE studies from Hong Kong Omicron wave, laboratory studies found neutralizing antibodies against Omicron after 2 doses of BNT162b2 or 2 doses of CoronaVac with undetectable titers (PMID: 35675370), yet these vaccine statuses still provide high protection against severity as predicted by antibody levels.

The reviewer suggests that protection from severe COVID-19 disease in the absence of detectable neutralising invalidates the analysis presented in this work. We feel it is important to stress that undetectable neutralising antibody titres does not mean the absence of neutralising antibodies. Instead, it reflects that most neutralisation assays are relatively insensitive (and cannot detect neutralisation titres below around 1 in 20). Indeed, our original analysis in 2021 found that in 5 / 7 phase 2 clinical trials for vaccines the 50% protective titre for protection from severe COVID-19 disease was below the limit of detection of neutralisation assays used. Analysis of antibody binding titres shows clearly that there is a continuum of antibody levels that extends well below the limit of detection in neutralisation assays (e.g. Wheatley et. al. *Nature Comms*, 2021), and that the detection threshold is an artefact of the assays themselves and not an intrinsic biological factor. Thus, we find it unsurprising to observe protection against severe disease with undetectable neutralisation titres.

We have clarified this section of the discussion:

[Our earlier] work found that while a level of neutralising antibodies equivalent to 20% of the GMT of early convalescent subjects (around 54 IU/ml) was associated with 50% protection from symptomatic infection, protection from severe infection was predicted to be achieved with a 6.5-fold lower titre (3.1% of convalescent, around 8 IU/ml)⁴. Unfortunately, the 50% protective level for symptomatic infection is close to the detection limit in most assays reported in the phase 1 / 2 vaccine studies. Similarly, the 50% protective titre from protection against severe infection is below the limit of detection in 5/7 of the reported assays⁴. This low sensitivity of neutralisation assays arises largely because of the relatively high serum dilutions used in most in vitro assays, with a serum dilution of 1:10 or 1:20 being the lowest tested in most cases^{34,35,36}. The relative insensitivity of the in vitro neutralisation assays has led to a perception of 'protection in the absence of neutralisation'³⁷. However, many subjects have clearly measurable antibody levels using antibody binding assays, even when these are not detectable by neutralisation assays³⁸. This shows that an 'undetectable' in vitro neutralisation titre is reflective of the limit of detection of the assay and does not necessarily indicate the absence of

neutralising antibodies. Thus, protection from severe SARS-CoV2 infection in the absence of detectable in vitro neutralisation, while perhaps not intuitive, is actually a clear prediction of the model⁴.

Last but not least, the authors replied by quoting the other reviewer as below:

We note that the comment from Reviewer 1 is in direct contrast to the comment from Reviewer 2 below, who noted that “researchers studying T-cell immunity have a tendency to talk up what they work on. They often make the point that T-cells protect against viral disease, including Covid-19. However, my own read of the Covid-19 vaccine literature is that the hard evidence supporting these claims is thin – almost to the point of non-existence.” These two very divergent views on the role of T-cells are indeed representative of the views of the scientific community at large (and the strong feelings of many on this topic), and therefore it is indeed important to address this question, which we have now done in the discussion.

In my opinion, this is a dangerous mindset in performing studies and drawing conclusion. Absent of evidence is not the evidence of absent. Ignoring contributions from other unmeasured immunizes may lead the field to overly focus on antibody while reducing exploration on other proactive mechanisms as stated here (PMID: 35324269). I am actually not studying T cell immunity, but an epidemiologist working with serological data and understand of the limitations of neutralizing antibodies in antigenically variable pathogens.

We thank reviewer 1 for their thoughts on the comments of reviewer 2 (which we had quoted). The reviewer has correctly noted that we are “*ignoring contributions from other unmeasured [mechanisms of immunity]*”. Instead, we focussed our attention on studies where robust data was available to either prove or disprove our hypothesis (in this case, the hypothesis we explored was whether neutralising antibodies are correlated with protection from severe disease).

The reviewer indicates that they “*understand the limitations of neutralizing antibodies*”, and that they are concerned that “*the field [is] overly focused on antibody while reducing exploration of other proactive mechanisms*”. We share the reviewer’s concerns that it is important to explore other mechanisms of immunity. Indeed, we have written extensively on the limitations of current studies of T cell immunity, and also proposed what we think are the best study designs to determine the T cell contribution (Kent et. al. *Nat Rev Imm*, 2022). In addition, we have published what we believe is the first demonstration that increased CD8⁺ T cell activation is associated with improved viral control after breakthrough infection (Koutsakos et. al. *Immunity in press* (2023) preprint on biorxiv). Thus, we share the reviewer’s opinion that other non-neutralising mechanisms of immunity need exploration and we are actively pursuing this through other work.

However, this manuscript does not address non-neutralising mechanisms of protection (other than to say they are important to explore). Instead, it aims to assess if there is any relationship between neutralising antibodies and protection from severe COVID-19. This does not exclude other correlates, but asks only whether neutralising antibodies are a correlate of severe outcomes. This work finds a strong correlation between predicted neutralising antibody titres and protection from severe COVID-19 disease, and that the data aligns well to a previously published model of the relationship between neutralisation and protection.

It is not clear what we can do in this manuscript to reduce the Reviewer’s perception that “*the field [is] overly focused on antibody*”. We cannot shy away from asking the question of whether neutralising

antibodies are correlated with protection from severe outcomes, because it might make *“the field to overly focus on antibody”*. Rather, by asking this question directly we could either have found an association or no association between neutralising antibodies and protection from severe outcomes. However, after analysing the available data, we are able to conclude that there is indeed a relationship and thus conclude that neutralising antibodies cannot be excluded as an important factor in protection from severe outcomes.

Reviewer #3

The authors have suitably revised the manuscript. Their search criteria and methodology and now easier to follow and much better explained.

We thank reviewer 3 for their comments.

Reviewer #4

All my concerns have been sufficiently addressed.

We thank reviewer 4 for their comment.